# Long QT syndrome-associated calmodulin variants disrupt the activity of the slowly activating delayed rectifier potassium channel

Liam McCormick[1,2] , Kirsty Wadmore[1], Amy Milburn[1], Nitika Gupta[1] , Rachael Morris[1], Marie Held[1] , Ohm Prakash[1] , Joseph Carr[1] , Richard Barrett-Jolley[3] , Caroline Dart[1] and Nordine Helassa[1]

[1]*Department of Biochemistry, Cell and Systems Biology, Institute of Systems, Molecular and Integrative Biology, Faculty of Health and Life Sciences, University of Liverpool, Liverpool, UK*
[2]*Manchester Centre for Genomic Medicine, North West Genomic Laboratory Hub, Saint Mary's Hospital, Manchester, UK*
[3]*Department of Musculoskeletal and Ageing Science, Institute of Life Course and Medical Sciences, Faculty of Health and Life Sciences, University of Liverpool, Liverpool, UK*

Handling Editors: Natalia Trayanova & Eleonora Grandi

The peer review history is available in the Supporting information section of this article (https://doi.org/10.1113/JP284994#support-information-section).

The Journal of Physiology

**Liam McCormick** is a Pre-Registration Clinical Scientist currently working in Solid Tumour genomics at the NHS North West Genomic Laboratory Hub. He earned his doctorate in Cellular and Molecular Physiology at the University of Liverpool before working as a postdoctoral research fellow in the Leeds Systems Physiology Lab at the University of Leeds. His work now involves providing molecular diagnostic and genomic testing to help guide the clinical management of cancer patients.

The Journal of Physiology

**Abstract** Calmodulin (CaM) is a highly conserved mediator of calcium ($Ca^{2+}$)-dependent signalling and modulates various cardiac ion channels. Genotyping has revealed several CaM mutations associated with long QT syndrome (LQTS). LQTS patients display prolonged ventricular recovery times (QT interval), increasing their risk of incurring life-threatening arrhythmic events. Loss-of-function mutations to Kv7.1 (which drives the slow delayed rectifier potassium current, IKs, a key ventricular repolarising current) are the largest contributor to congenital LQTS (>50% of cases). CaM modulates Kv7.1 to produce a $Ca^{2+}$-sensitive IKs, but little is known about the consequences of LQTS-associated CaM mutations on Kv7.1 function. Here, we present novel data characterising the biophysical and modulatory properties of three LQTS-associated CaM variants (D95V, N97I and D131H). We showed that mutations induced structural alterations in CaM and reduced affinity for Kv7.1, when compared with wild-type (WT). Using HEK293T cells expressing Kv7.1 channel subunits (KCNQ1/KCNE1) and patch-clamp electrophysiology, we demonstrated that LQTS-associated CaM variants reduced current density at systolic $Ca^{2+}$ concentrations (1 $\mu$M), revealing a direct QT-prolonging modulatory effect. Our data highlight for the first time that LQTS-associated perturbations to CaM's structure impede complex formation with Kv7.1 and subsequently result in reduced IKs. This provides a novel mechanistic insight into how the perturbed structure–function relationship of CaM variants contributes to the LQTS phenotype.

(Received 8 May 2023; accepted after revision 21 June 2023; first published online 30 June 2023)

**Corresponding author** Nordine Helassa: Department of Biochemistry, Cell and Systems Biology, Institute of Systems, Molecular and Integrative Biology, Faculty of Health and Life Sciences, University of Liverpool, Liverpool L69 3BX, UK. Email: nhelassa@liverpool.ac.uk

**Abstract figure legend** LQTS-CaM proteins displayed structural differences, with CaM variants undergoing less $Ca^{2+}$-induced conformational change when compared with CaM-WT. These alternative conformations contribute to a reduced binding affinity of CaM to the C-terminus of Kv7.1 (KCNQ1), more particularly in the presence of $Ca^{2+}$. In cells, this translates to a reduction in IKs for LQTS-associated CaM variants, and subsequently the prolongation of the ventricular action potential (QT interval). These findings highlight how the perturbed structure–function relationship of CaM variants and Kv7.1 contributes to the LQTS phenotype.

## Key points

- Calmodulin (CaM) is a ubiquitous, highly conserved calcium ($Ca^{2+}$) sensor playing a key role in cardiac muscle contraction.
- Genotyping has revealed several CaM mutations associated with long QT syndrome (LQTS), a life-threatening cardiac arrhythmia syndrome.
- LQTS-associated CaM variants (D95V, N97I and D131H) induced structural alterations, altered binding to Kv7.1 and reduced IKs.
- Our data provide a novel mechanistic insight into how the perturbed structure–function relationship of CaM variants contributes to the LQTS phenotype.

## Introduction

Long QT syndrome (LQTS) is an inherited life-threatening cardiac disorder with a predicted incidence of approximately 1 in 2000 (Schwartz et al., 2009; Schwartz et al., 2012). It is characterised by a prolonged QT interval on an electrocardiogram, caused by loss-of-function or gain-of-function mutations in cardiac ion channels that lead to prolongation of ventricular repolarisation (Gemma et al., 2011; Shimizu & Antzelevitch, 1999). Approximately 50% of LQTS cases

present asymptomatically, making patients unaware of their underlying condition (Cohagan & Brandis, 2021). Patients suffering with LQTS may experience rapid, chaotic heartbeats that can result in syncope, seizures, ventricular arrhythmias and sudden cardiac death (Baskar & Aziz, 2015; Schwartz et al., 2001). Over 15 congenital subtypes of LQTS have been described, many of which are associated with ventricular ion channel dysfunction (Lorca et al., 2022; Schwartz et al., 2020; Song & Shou, 2012). Recently, mutations in the highly conserved calcium ($Ca^{2+}$)-sensing protein calmodulin (CaM) have

been associated with LQTS (Boczek et al., 2016; Chaix et al., 2016; Chazin & Johnson, 2020; Crotti et al., 2013; Crotti et al., 2019; Fujita et al., 2019; Hussey et al., 2023; Jensen et al., 2018; Jimenez-Jaimez et al., 2016; Makita et al., 2014; Reed et al., 2015; Schwartz et al., 2020; Wren et al., 2019). However, the underlying molecular mechanisms which promote arrhythmogenic phenotypes in patients harbouring these CaM mutations remain elusive.

CaM is a highly conserved, ubiquitous $Ca^{2+}$-sensing protein which serves as a central mediator of $Ca^{2+}$-dependent signalling. It is encoded by three independent genes (*CALM1*, *CALM2* and *CALM3*) which all translate an identical 148 amino acid long protein (Fischer et al., 1988; Friedberg & Rhoads, 2001a, b; Halling et al., 2016). CaM contains four EF-hand motifs which can bind up to four $Ca^{2+}$ ions, EF-hands 1 and 2 are found in the N-lobe while EF-hands 3 and 4 are in the C-lobe. Both lobes are tethered to each other in a 'dumbbell-like' manner by a flexible $\alpha$-helix linker, this allows CaM to conformationally adapt to embrace many target proteins (Ikura & Ames, 2006; Shimoyama & Takeda-Shitaka, 2017; Zhang et al., 2012). Furthermore, the isolated, yet physically coupled globular domains of CaM permit a dynamic interplay with target binding where target affinity of one lobe can be modulated depending on the complex formed at the neighbouring lobe (Sondergaard et al., 2017). Upon binding to $Ca^{2+}$, CaM undergoes significant conformational change and adopts a more open conformation, exposing otherwise buried hydrophobic patches. This reversible conformational transition permits $Ca^{2+}$-dependent interaction with, and modulation of, a wide range of targets. CaM is therefore able to dynamically translate changes in intracellular $[Ca^{2+}]$ to mediate a myriad of cellular processes, including cardiac muscle contraction. In cardiomyocytes, CaM regulates many of the central proteins involved in excitation–contraction coupling such as the L-type voltage-gated $Ca^{2+}$ channel (Cav1.2), the cardiac ryanodine receptor (RyR2), the voltage-gated $Na^+$ channel (Nav1.5) and the voltage-gated $K^+$ channel (Kv7.1) (Balshaw et al., 2002; Brohus et al., 2019; Gabelli et al., 2014; Ghosh et al., 2006; Kang et al., 2020; Kang et al., 2021; Xu & Meissner, 2004; Yamaguchi et al., 2003; Zuhlke et al., 1999).

The slow delayed rectifier potassium current (IKs) is a key ventricular repolarising current generated by Kv7.1 (KCNQ1) when modulated by the accessory sub-unit mink (KCNE1). Loss-of-function mutations within Kv7.1 are the most common disease aetiology of LQTS, accounting for over 50% of cases (Schwartz et al., 2012). CaM can bind and modulate Kv7.1 through interaction with two C-terminal domains (helix A, HA and helix B, HB) in both in the absence and presence of $Ca^{2+}$ (Sachyani et al., 2014; Sun & MacKinnon,

2017, 2020; Wiener et al., 2008; Yus-Najera et al., 2002). CaM is considered an obligate subunit of Kv7.1, modulating tetramer assembly, channel folding and membrane trafficking (Ghosh et al., 2006; Shamgar et al., 2006). $Ca^{2+}$-dependent regulation of Kv7.1 by CaM potentiates current generation, permitting efficient repolarisation of the myocyte membrane in response to elevated cytoplasmic $Ca^{2+}$ concentration ($[Ca^{2+}]_{cyt}$) (Bai et al., 2005; Bartos et al., 2017; Nitta et al., 1994; Shamgar et al., 2006; Tobelaim et al., 2017; Tohse, 1990). Because Kv7.1 is unable to bind or detect $Ca^{2+}$ alone (Ghosh et al., 2006), CaM modulation of Kv7.1 allows fine tuning of IKs in respect to depolarising stimuli ($Ca^{2+}$), for an appropriately timed membrane repolarisation and a feedback-regulated action potential duration. Perturbation of this CaM-facilitated communication between Kv7.1 and $[Ca^{2+}]_{cyt}$ is prone to cause ventricular arrhythmia either through an abnormal enhanced IKs (promoting short QT syndrome) or diminished IKs generation (promoting LQTS) (Bartos et al., 2017; Nitta et al., 1994; Rudic et al., 2014; Tohse, 1990). Previous studies have demonstrated that Kv7.1 mutations which alter CaM interactions perturb channel function (Ghosh et al., 2006; Gonzalez-Garrido et al., 2021; Sachyani et al., 2014; Shamgar et al., 2006; Tobelaim et al., 2017; Yang et al., 2009; Zhou et al., 2016). LQTS-CaM mutants have displayed a range of disturbed regulatory effects across cardiac targets, including reduced $Ca^{2+}$-dependent inactivation of Cav1.2 (Boczek et al., 2016; Gomez-Hurtado et al., 2016; Limpitikul et al., 2014; Limpitikul et al., 2017; Pipilas et al., 2016; Prakash et al., 2023; Rocchetti et al., 2017; Yamamoto et al., 2017; Yin et al., 2014), altered inhibition of RyR2 (Gomez-Hurtado et al., 2016; Hwang et al., 2014; Nomikos et al., 2014; Prakash et al., 2021; Sondergaard et al., 2015; Sondergaard et al., 2017; Vassilakopoulou et al., 2015) and pathological dysregulation of CaMKII$\delta$ (Berchtold et al., 2016; Limpitikul et al., 2017; Prakash et al., 2023). However, whether these CaM mutations alter interactions with Kv7.1 and infer altered modulation of IKs, remains largely unexplored.

In this paper, we investigate three LQTS-associated CaM variants: D95V, N97I and D131H. These variants have missense mutations within the C-lobe of CaM, specifically at residues which directly coordinate $Ca^{2+}$ binding (Fig. 1). D95V and N97I are located in EF-hand III, while D131H is in EF-hand IV. The consequences of these variants have previously been clinically described in patients presenting with *de novo* CaM (*CALM2*) mutations (Crotti et al., 2013; Makita et al., 2014; Pipilas et al., 2016). The clinical presentations of all mutations were highly pathogenic and resulted in severe and recurrent episodes of life-threatening arrythmias. The perturbed $Ca^{2+}$-binding ability of D95V, N97I and D131H CaM variants is a likely driver of disease pathogenesis

(Crotti et al., 2013; Makita et al., 2014; Pipilas et al., 2016). Previous studies highlight the difficulty in predicting how perturbed $Ca^{2+}$ binding in arrhythmia-associated CaM variants translates to defective target modulation (Chazin & Johnson, 2020; Hussey et al., 2023; Jensen et al., 2018). We employed a multidisciplinary approach to gain novel insight into the molecular aetiology of how LQTS-associated CaM mutants contribute to arrythmia through impaired Kv7.1 function. We demonstrate that LQTS-associated CaM mutations reduce IKs, most likely through altered CaM structure and impaired interaction with Kv7.1. The presented data aid in elucidating the contributions of LQTS-associated CaM mutants in electrical disease of the heart and how they may exacerbate the most common mechanism of LQTS, perturbed IKs generation.

## Methods

### Molecular biology

**For biophysical and structural biology experiments.** The sequence of human wild-type (WT) CaM was subcloned into the pE-SUMOPro-Kan vector (LifeSensors, USA) as previously described (Prakash et al., 2021, 2023). A series of site-directed mutagenesis (SDM) reactions were performed using the QuikChangeII kit (Agilent Technologies, USA), according to the manufacturer's recommendation, in order to generate LQTS-associated CaM mutants D95V, N97I and D131H. Primers used are presented in Table 1 (SDM primers).

**For electrophysiology experiments.** DNA corresponding to CaM variants (from the pE-SUMOPro-Kan constructs), KCNQ1 and KCNE1 (Addgene plasmid 53 048, 53 050, gifts from Michael Sanguinetti) (Sanguinetti et al., 1996) was subcloned into pHIV-IRES-dTomato or pHIV-IRES-EGFP (Addgene plasmid 21 374, 21 373, gifts from Bryan Welm) using Gibson Assembly (NEBuilder HiFi, New England Biolabs), according to the manufacturer's guidelines. The primers used are presented in Table 1 (IRES primers). In these constructs, proteins of interest and a fluorescent marker (dTomato or EGFP) were co-expressed under the control of the same promoter, as two distinct proteins and not as fusion proteins.

**For flow cytometry experiments.** The trafficking assay requires KCNQ1-BBS, a modified KCNQ1 construct whereby a 13-residue bungarotoxin-binding site (BBS) was introduced into the extracellular S1–S2 loop to allow channel detection at the cell surface through a cell-impermeable, fluorescent $\alpha$-bungarotoxin (BTX) conjugate. KCNQ1-BBS (gift from Henry M. Colecraft) (Aromolaran et al., 2014), ECFP (Addgene plasmid 70 104, gift from Harald Sitte) (Sucic et al., 2010) and the pHIV backbone (Addgene plasmid 21 373, gift from Bryan Welm) were assembled in a single reaction via Gibson Assembly (NEBuilder HiFi, New England Biolabs) according to the manufacturer's guidelines. The primers were designed to remove the IRES-EGFP sequence, allowing the generation of pHIV-ECFP-KCNQ1-BBS (Table 1). The regulatory subunit KCNE1/minK was a gift from Michael Sanguinetti (Addgene plasmid 53 050) (Sanguinetti et al., 1996). For CaM constructs, the CaM variants were subcloned from the pGEX-6P-1 vector (Lian et al., 2014) into pEGFP-N1 by restriction-ligation (KpnI/BamHI) using primers displayed in Table 1. A series of SDM reactions were performed following the QuikChangeII protocol (Agilent Technologies) as

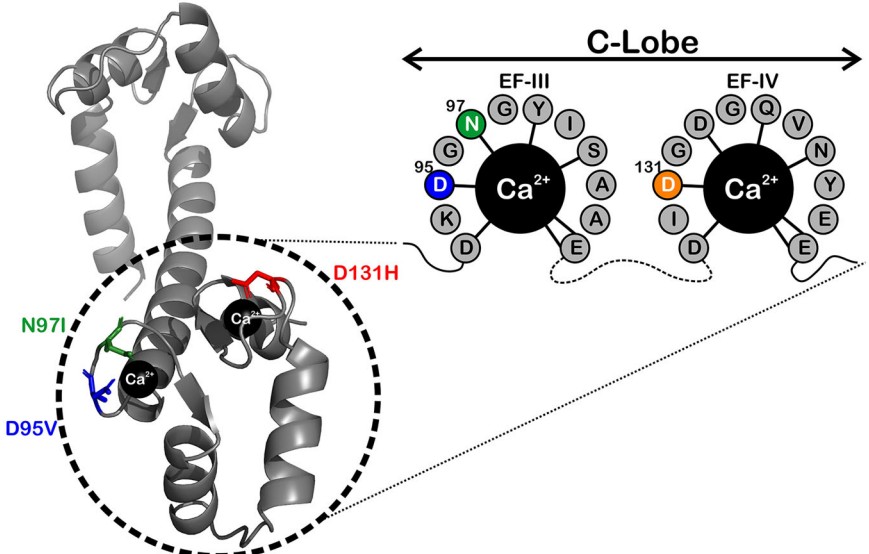

**Figure 1. Representation of $Ca^{2+}$/CaM highlighting LQTS-associated mutations**
*Left* shows the location of the mutations in the crystal structure of $Ca^{2+}$/CaM (PDB: 1CLL). *Right* illustrates their location within the C-lobe, all of which occur within residues which directly coordinate $Ca^{2+}$, as depicted by black lines. Mutants D95V and N97I are located in the third $Ca^{2+}$-binding site of CaM (EF-hand III), whereas D131H is found in the fourth EF-hand (EF-hand IV).

**Table 1. List of primers used in this study**

| | Primer name | Primer sequence (5′−3′) |
|---|---|---|
| SDM | D95V FW | ATTCCGTGTGTTTGATAAGGTTGGCAATGGCTATATTAGTG |
| | D95 RV | CACTAATATAGCCATTGCCAACCTTATCAAACACACGGAAT |
| | N97I FW | TCCGTGTGTTTGATAAGGATGGCATTGGCTATATTAGTG |
| | N97I RV | CACTAATATAGCCAATGCCATCCTTATCAAACACACGGA |
| | D131H FW | ATGATCAGGGAAGCAGATATTCATGGTGATGGTCAAG |
| | D131H RV | CTTGACCATCACCATGAATATCTGCTTCCCTGATCAT |
| CaM-IRES-EGFP | pHIV FW | TAGGATCCGCCCCTCTCCC |
| | pHIV RV | CGCTCACGACACCTGAAATG |
| | CaM FW | CATTTCAGGTGTCGTGAGCGATGGCTGACCAACTGACTG |
| | CaM RV | GAGGGAGAGGGGCGGATCCTTCACTTTGCTGTCATCATTTG |
| KCNQ1-IRES-dTomato | pHIV FW | AGGATCCGCCCCTCTCCC |
| | pHIV RV | GCGGCCGCTCACGACACC |
| | KCNQ1 FW | CAGGTGTCGTGAGCGGCCGCATGGCCGCGGCCTCCTCC |
| | KCNQ1 RV | GAGGGAGAGGGGCGGATCCTTCAGGACCCCTCATCGGGGC |
| KCNE1-IRES-dTomato | pHIV FW | TAGGATCCGCCCCTCTCCC |
| | pHIV RV | CGCTCACGACACCTGAAATG |
| | KCNE1 FW | CAGGTGTCGTGAGCGGCCGCATGATCCTGTCTAACACCACAG |
| | KCNE1 RV | GAGGGAGAGGGGCGGATCCTTCATGGGGAAGGCTTCGTC |
| ECFP-KCNQ1-BBS | pHIV FW | AGCGGCCGCATCGATACC |
| | pHIV RV | TTCTAGAATAATCAATAGTTAACTCAGCGGCC |
| | ECFP FW | AACTATTGATTATTCTAGAAATGGTGAGCAAGGGCGAG |
| | ECFP RV | CCGCGGCCATCTTGTACAGCTCGTCCATGC |
| | KCNQ1-BBS FW | GCTGTACAAGATGGCCGCGGCCTCCTCC |
| | KCNQ1-BBS RV | ACGGTATCGATGCGGCCGCTTCAGGACCCCTCATCGGGGC |
| CaM-EGFP | CaM FW | CAATGGTACCATGGCTGACC |
| | CaM RV | GGTAGGATCCATCTTTGCTGTCATC |

described above, to generate LQTS-associated CaM constructs. These CaM constructs were fused to EGFP by their C-terminal domain.

All molecular constructs were verified by DNA sequencing (MRC PPU, University of Dundee, UK).

### Whole-cell patch-clamp electrophysiology

HEK293T cells were cultured at 37°C, 5% $CO_2$ in Dulbecco's Modified Eagle Medium (DMEM) GlutaMAX (ThermoFisher) supplemented with 10% heat-inactivated fetal bovine serum (ThermoFisher), 1% (v/v) MEM Non-Essential Amino Acids (ThermoFisher) and 100 U/ml penicillin-streptomycin (ThermoFisher). Cells ($0.3 \times 10^6$) were plated in standard six-well plates and co-transfected with CaM-IRES-EGFP, KCNQ1-IRES-dTomato and KCNE1-IRES-dTomato using Lipofectamine2000 (2:1:1 molar ratio), following the manufacturer's recommendations. Cells were harvested 48 h post-transfection with 0.05% Trypsin-EDTA (Gibco), gently pelleted and washed in PBS, pH 7.4 (Gibco) for cellular experiments.

Recordings were collected using an Axon Axopatch 200B amplifier (Molecular Devices) in whole-cell voltage-clamp configuration. Currents were filtered at 2 kHz and digitised at 10 kHz using an Axon Digidata 1320A interface (Molecular Devices). All recordings were performed at room temperature. The bath solution consisted of 140 mM NaCl, 11 mM glucose, 5.5 mM $Na^+$-HEPES, 4 mM KCl, 1.8 mM $CaCl_2$ and 1.2 mM $MgCl_2$, at pH 7.4. The pipette solution consisted of 130 mM KCl, 10 mM HEPES, 5 mM EGTA, 1 mM $Na^+$-ATP, 1 mM $MgCl_2$ supplemented with 0 or 1 $\mu$M free $[Ca^{2+}]$, as calculated using Maxchelator (Bers et al., 2010). Patch pipettes were pulled from thin-walled borosilicate glass microelectrodes (outside diameter: 1.50 mm, inside diameter: 1.17 mm) (Harvard Apparatus) and fire-polished to a resistance of approximately 5.0 MΩ.

HEK293T cells transiently transfected with CaM-IRES-EGFP, KCNQ1-IRES-dTomato and KCNE1-IRES-dTomato were added dropwise into the perfusion bath and allowed to settle onto the glass-bottomed chamber. CaM-expressing cells were identified by EGFP fluorescence (Nikon Eclipse TE200 inverted microscope with epifluorescence attachment). The IKs activation protocol consisted of 4 s voltage steps ranging from −60 to +100 mV (20 mV increments) from a holding potential of −80 mV, followed by

a 3 s repolarisation step at −40 mV. Half maximal activation ($V_{1/2}$) was calculated by fitting normalised peak conductance using the Boltzmann sigmoid equation. All data analysis was performed using the Axon pClamp software package (version 10.7.0.3, Molecular Devices) and GraphPad Prism.

### Recombinant protein expression and purification of CaM variants

CaM proteins were recombinantly expressed in *Escherichia coli* BL21 (DE3) STAR, purified using affinity and size-exclusion chromatography, as previously described (Prakash et al., 2021, 2023). For nucleic magnetic resonance (NMR) experiments, uniformly isotopic-labelled CaM proteins were grown in minimal medium containing 88 mM $Na_2HPO4$, 55 mM $KH_2PO_4$, 30 $\mu$M thiamine-HCl, 136 $\mu$M $CaCl_2$, 1 mM $MgSO_4$, 19 mM $^{15}NH_4Cl$ (Cambridge Isotope Laboratories) and 22 mM glucose. Recombinant overexpression and purification were performed as described above. Isotopically labelled reagents were purchased from Cambridge Isotope Laboratories. Unlabelled and isotopically labelled CaM protein purity was determined by SDS-PAGE (NuPAGE 4–12% Bis-Tris, Life Technologies) and Coomassie staining (InstantBlue, Abcam). The final protein concentration was determined by measuring absorbance at 280 nm using a DS-11+ spectrophotometer (DeNovix) and the molar extinction coefficient calculated from the amino acid composition (ExPASy/ProtParam program) ($\varepsilon_0$ (CaM) = 2980 $M^{-1}$ $cm^{-1}$) (Gasteiger et al., 2005).

### Secondary structure content and thermal stability

**Secondary structure content.** Far-UV spectra (180–260 nm) were collected at 20°C in a 0.1 cm path length quartz cell using a Jasco J-1100 circular dichroism spectrometer equipped with a Jasco MCB-100 mini circulation bath for temperature control. CaM (10 $\mu$M) spectra were measured in 2 mM HEPES, pH 7.5 supplemented with either 1 mM EGTA (pH 7.5) or 5 mM $CaCl_2$ for $Ca^{2+}$-free (apo) and $Ca^{2+}$-bound experiments, respectively. For each sample, three scans were averaged (scan rate of 100 nm/min). After buffer subtraction, data were normalised to mean residual ellipticity and secondary structure content was calculated using the CDSSTR prediction programme (Dichroweb, reference dataset 7) (Whitmore & Wallace, 2004, 2008).

**Thermal stability.** Sensitivity of apo-CaM (10 $\mu$M) to temperature was assessed by decrease in $\alpha$-helical content measured by circular dichroism at 222 nm. Temperature ranged from 15°C to 90°C in 1°C increments, with a ramp increase rate of 1°C/min and a 180 s equilibration period between recordings. Data were normalised and fitted to the Boltzmann sigmoid equation (GraphPad Prism) to derive the melting temperature of CaM ($T_m$).

### Susceptibility to trypsin digestion

Sensitivity of CaM to trypsin hydrolysis was assessed by SDS-PAGE and densitometry analysis. Purified CaM proteins (5 $\mu$M) were incubated with trypsin for 30 min at 37°C in 25 mM HEPES, 100 mM NaCl, pH 7.5 supplemented with trypsin at 0–10 mg/ml in apo conditions (10 mM EGTA) and 0–30 mg/ml in $Ca^{2+}$-bound conditions (5 mM $CaCl_2$). Reactions were rapidly terminated by the addition of SDS-containing sample buffer and heating at 95°C for 10 min. Proteins were separated by SDS-PAGE (NuPAGE 4–12% Bis-Tris, Life Technologies) and stained with InstantBlue (Abcam). Images were obtained on a ChemiDoc XRS+ trans-illuminator (Bio-Rad) and the fraction of intact CaM was quantified by densitometry using Fiji software (Schindelin et al., 2012).

### Kv7.1 peptides

Peptides were designed according to the CaM-binding domains within the C-terminus of KCNQ1, namely helix A (HA, residues 370−389) and helix B (HB, residues 507−536) (Yus-Najera et al., 2002). Peptides were chemically synthesised and HPLC purified (>95% purity) (GenicBio).

Kv7.1-$HA_{370-389}$: AAASLIQTAWRCYAAENPDS

Kv7.1-$HB_{507-536}$: REHHRATIKVIRRMQYFVAKKKFQ QARKPY

Peptide concentration was determined by measuring absorbance at 280 nm using a DS-11+ spectrophotometer (DeNovix) and molar extinction coefficients calculated from the amino acid composition (ExPASy/ProtParam program). $\varepsilon_0$ (Kv7.1-$HA_{370-389}$) = 6990 $M^{-1}$ $cm^{-1}$ and $\varepsilon_0$ (Kv7.1-$HB_{507-536}$) = 2980 $M^{-1}$ $cm^{-1}$ (Gasteiger et al., 2005).

### Nucleic magnetic resonance spectroscopy

NMR spectra were collected at 30°C (303 K) on an Avance III 800 MHz or Ascend 700 MHz spectrometer equipped with [$^1$H, $^{15}$N]-cryoprobes (Bruker). $^1$H-$^{15}$N HSQC spectra were acquired for $^{15}$N-labelled CaM variants (50–100 $\mu$M) in 20 mM $Na^+$-HEPES (pH 7.5), 50 mM NaCl, 1 mM $CaCl_2$, 10% (v/v) $D_2O$ supplemented with 1 mM EGTA (apo-CaM) or 1 mM $CaCl_2$ ($Ca^{2+}$-CaM). Backbone amino acid assignments for CaM variants were transferred from previous work (Prakash et al., 2021) and chemical shift differences were expressed in

ppm as $\Delta\delta = [(\Delta H)^2 + (0.15\Delta N)^2]^{1/2}$. To determine the structural changes induced by CaM interaction to Kv7.1, $^1H$-$^{15}N$ HSQC spectra were recorded after stepwise addition of Kv7.1-HB$_{507\text{-}536}$ peptide to the protein sample to achieve 1:1 peptide:protein molar ratio. Raw data were processed using Bruker TopSpin software. Resonance peaks were analysed and assigned using CcpNmr software (Vranken et al., 2005).

### Isothermal titration calorimetry (ITC)

Experiments were performed in 50 mM Na$^+$-HEPES, 100 mM KCl, 2 mM MgCl$_2$ (pH 7.5) supplemented with either 1 mM EGTA or 5 mM CaCl$_2$ for Ca$^{2+}$-independent or -dependent interactions, respectively. Kv7.1-HA$_{370\text{-}389}$ or Kv7.1-HB$_{507\text{-}536}$ peptides were titrated against CaM proteins across 20 injections (2 $\mu$l each) lasting 4 s with a 180 s grace period between each injection. Peptide was typically titrated into the cell at a concentration 10–20 times higher than CaM ([CaM] $\sim$ 50 $\mu$M). All titrations were performed using MicroCal iTC200 and automated PEAQ-ITC systems (Malvern Panalytical) at 25°C with continuous stirring at 800 rpm. Data were processed using MicroCal PEAQ-ITC software and fitted according to a one-site or two-site binding model to determine binding characteristics (affinity, stoichiometry, enthalpy change, entropy change and Gibbs free energy).

### Flow cytometry

Kv7.1 cell surface density was determined by flow cytometry in live HEK293T cells expressing ECFP-KCNQ1-BBS, KCNE1 and CaM-EGFP variants, following previously described protocols (Aromolaran et al., 2014). Briefly, cells ($0.3 \times 10^6$) were plated in standard six-well plates and co-transfected with ECFP-KCNQ1-BBS, KCNE1 and CaM-EGFP variants using Lipofectamine2000 (1:1:1 molar ratio), following the manufacturer's recommendations. Cells were gently washed with ice cold PBS containing 1 mм CaCl$_2$ and 0.5 mм MgCl$_2$ (pH 7.4). Post-transfection (48 h), cells were blocked in DMEM/3% bovine serum albumin on ice for 30 min and then incubated with 1 $\mu$M Alexa Fluor 647 conjugated $\alpha$-bungarotoxin (ThermoFisher) on ice for 1 h in the dark. Cells were washed three times with PBS (containing Ca$^{2+}$ and Mg$^{2+}$) and harvested with 0.05% Trypsin-EDTA (ThermoFisher). Cells were resuspended in normal PBS and assayed using a BD FACSCanto II flow cytometer (BD Biosciences). ECFP- and EGFP-tagged proteins were excited with violet (405 nm) and blue (488 nm) lasers, respectively, and Alexa Fluor 647 was excited with a red (633 nm) laser. Fluorescence signals from flow cytometry were analysed using the BD FACSDiva 9.0 software. For each group of experiments, live cells were discriminated from heterogenous populations by size through comparison of cell area via side and forward scatter (SSC-A and FSC-A). Single cells were gated from doublets through comparison of forward scatter height and area (FSC-H and FSC-A). Isochronal untransfected and single colour controls were used to manually set the threshold and gain setting for each fluorophore.

After selecting for EGFP-positive cells (CaM), Kv7.1 surface density was calculated as:

$$\frac{\left(\text{ECPF and Alexa Fluor 647 positive cells}\right) - \left(\text{Alexa Fluor 647 only positive cells}\right)}{\left(\text{ECFP positive cells}\right)} \tag{1}$$

### Mathematical modelling

To model the likely action potential duration effects of direct CaM blocking of IKs we deployed the O'Hara–Rudy (2017) ventricular myocyte model (O'Hara et al., 2011) using the CellML implementation (Lloyd et al., 2008). The model is available at https://models.cellml. org/e/5a0/ohara_rudy_cipa_v1_2017.cellml/view. We made a simple addition to the O'Hara–Rudy model in terms of a direct CaM-IKs interaction where CaM concentration blocked IKs according to the Hill equation (eqn 2):

$\frac{[c]}{([c]+Kb)}$, where [c] is the CaM concentration specified elsewhere in the model and Kb is an arbitrary constant leading to the experimentally observed block in peak IKs density.

### Statistical analysis

Statistical analyses were performed using GraphPad Prism version 9. Number of replicates and type of statistical tests are indicated in the figure legends. $P$ value $<0.05$ was considered statistically significant. $P$ values are represented by asterisks with $^*P < 0.05$, $^{**}P < 0.01$, $^{***}P < 0.001$, $^{****}P < 0.0001$.

## Results

### Arrhythmogenic CaM variants reduce IKs densities and impair voltage-dependent activation at resting (100 nM) and high (1 μM) Ca²⁺ levels

To investigate the effect of the LQTS-associated CaM variants on IKs, whole-cell patch-clamp electrophysiology was performed using HEK-293T cells co-transfected with CaM, KCNQ1 and KCNE1. At resting intracellular Ca²⁺ levels (100 nM), we observed a significant decrease in IKs densities (Fig. 2*A*, *B*) and disrupted activation kinetics when channels were modulated by LQTS-CaM variants (Fig. 2*C*). Current densities at +100 mV were significantly reduced for CaM-D95V, CaM-N97I and CaM-D131H, when compared with CaM-WT (Fig. 2*A*, *B* and Table 2). The voltage at which 50% of the channels were active ($V_{1/2}$) was reduced by up to threefold for CaM-D95V and CaM-D131H, when compared with CaM-WT (Fig. 2*C*, Table 2).

At systolic levels of intracellular Ca²⁺ (1 μM), reduced current densities and impaired activation kinetics were observed for CaM-D95V and CaM-N97I (Fig. 3). Current densities at +100 mV were reduced for CaM-D95V and CaM-N97I, when compared with CaM-WT (Fig. 3*A*, *B* and Table 2). CaM-D131H remained unchanged. The voltage at which 50% of the channels were active ($V_{1/2}$) were reduced for CaM-D95V and CaM-D131H, while modulation by CaM-N97I remained similar to CaM-WT (Fig. 3*C*, Table 2). Together, this indicates that LQTS-associated CaM variants affect Kv7.1 activity at both resting and high Ca²⁺ concentrations.

### Trafficking of KCNQ1 channels to the plasma membrane is not altered by the LQTS-CaM variants

A reduction in whole-cell currents can be caused by changes in channel behaviour or by a change in the number of active channels at the cell surface. Using flow

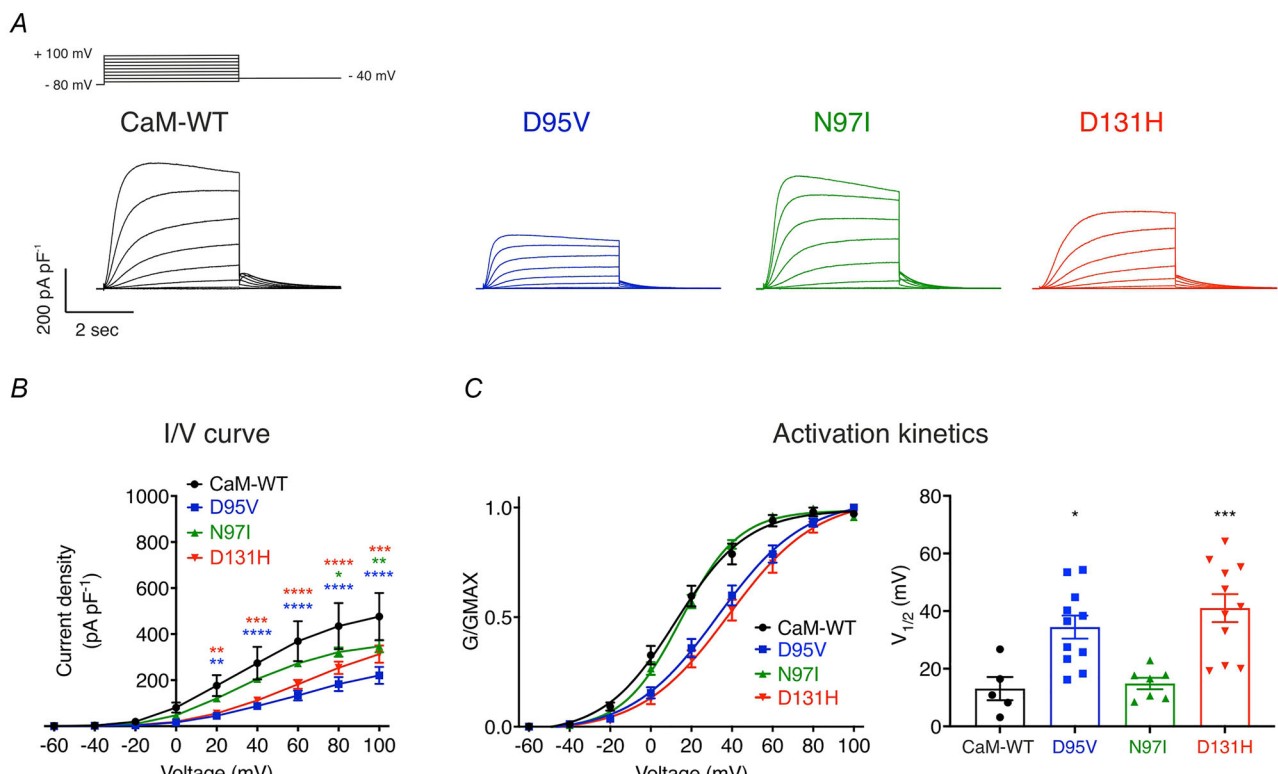

**Figure 2. LQTS-associated CaM mutations decrease IKs densities and reduce voltage sensitivity at resting intracellular [Ca²⁺] (100 nM)**
*A*, representative traces from HEK293T cells transiently transfected with KCNQ1, KCNE1 and CaM variants. Currents were obtained in whole-cell voltage-clamp configuration by holding cells at −80 mV and stepping for 4 s from −60 mV to +100 mV in 20 mV increments, followed by a repolarising step at −40 mV. *B*, current–voltage (I/V) relationships of IKs currents modulated by CaM. Differences between groups were determined using a two-way ANOVA with Dunnett's multiple comparisons tests. *C*, activation kinetics. (Left panel) mean ± s.e.m. Channel conductance, G, normalised to peak conductance, Gmax, to give mean activation/activation curves. (Right panel) mean ± s.e.m. Half maximal activation voltages, $V_{1/2}$, calculated from individual curves fitted using the Boltzmann equation. Differences between groups were determined using a one-way ANOVA with Dunnett's multiple comparisons tests.

**Table 2. Summary of IKs densities and voltage sensitivity at resting (100 nM) and high (1 $\mu$M) intracellular Ca$^{2+}$ levels. Data are means ± s.e.m. Statistical significance was determined using two-way ANOVA with Dunnett's multiple comparisons tests (for current densities) and one-way ANOVA with Dunnett's multiple comparisons tests (for V$_{1/2}$ activation)**

| | Variant | Current density at +100 mV (pA/pF) | *P* value (number of replicates) | V$_{1/2}$ activation (mV) | *P* value (number of replicates) |
|---|---|---|---|---|---|
| Resting Ca$^{2+}$ levels (100 nM) | CaM-WT | 476.1 ± 102.4 | (5) | 13.1 ± 4.0 | (5) |
| | D95V | 220.5 ± 37.0 | <0.0001 (11) | 34.4 ± 4.0 | 0.0102 (11) |
| | N97I | 347.0 ± 25.8 | 0.0065 (8) | 14.9 ± 2.0 | 0.9878 (7) |
| | D131H | 312.9 ± 37.2 | 0.0001 (11) | 41.0 ± 4.9 | 0.0008 (11) |
| High Ca$^{2+}$ levels (1 $\mu$M) | CaM-WT | 505.4 ± 76.0 | (9) | 22.2 ± 2.9 | (9) |
| | D95V | 341.1 ± 38.2 | 0.0002 (12) | 31.6 ± 2.7 | 0.0331 (12) |
| | N97I | 331.9 ± 41.2 | <0.0001 (11) | 22.5 ± 2.1 | 0.9996 (11) |
| | D131H | 415.9 ± 63.0 | 0.1118 (8) | 32.1 ± 2.6 | 0.0427 (8) |

cytometry, we quantitatively assessed the Kv7.1 (KCNQ1) relative surface density in live cells to determine whether trafficking of the channel was impaired in the presence of the CaM variants. KCNQ1 was expressed as an ECFP fusion and contained a 13-residue high-affinity $\alpha$-BBS in its extracellular S1–S2 loop to allow labelling with extracellularly applied $\alpha$-bungarotoxin-Alexa Fluor 647. Therefore, ECFP levels represented the total Kv7.1 expressed in cells, while ECFP+Alexa Fluor 647 levels indicated Kv7.1 channels at the surface (Fig. 4*A*). Based on this method developed by Colecraft's laboratory (Aromolaran et al., 2014), we observed that KCNQ1 surface density (ECFP+Alexa Fluor 647 labelled) was 35.0 ± 1.2% (of total KCNQ1 channels, ECFP only, $n = 5$) and that LQTS-associated CaM variants did not significantly alter the percentage of channels at the plasma membrane. Kv7.1 surface density was 30.8 ± 2.6% for D95V ($n = 5$, $P = 0.3140$), 30.4 ± 1.7% ($n = 5$, $P = 0.2502$) for N97I and 35.4 ± 1.9% for D131H ($n = 5$, $P = 0.9976$) (Fig. 4).

### LQTS-associated mutations induce changes in CaM structure

Secondary structure content was investigated by far-UV circular dichroism (Fig. 5). In the absence of Ca$^{2+}$ (apo conditions), the $\alpha$-helical content was decreased for CaM-N97I (32 ± 2%, $n = 5$) and increased for D131H (39 ± 1%, $n = 5$), when compared with CaM-WT (36 ± 1%, $n = 5$). In the presence of Ca$^{2+}$, we observed a characteristic increase in $\alpha$-helical content for all variants (from 36 ± 1% to 59 ± 1% for CaM-WT, $n = 5$; from 38 ± 1% to 58 ± 1% for D95V, $n = 5$; from 32 ± 2% to 59 ± 1% for N97I, $n = 5$), except for D131H (from 39 ± 1% to 42 ± 1%, $n = 5$).

Using $^1$H-$^{15}$N HSQC NMR, we investigated structural differences between the CaM variants in Ca$^{2+}$-bound conditions (Fig. 6). Spectra showed that LQTS-associated variants exhibited distinct spectra when compared with CaM-WT (Fig. 6*A*). Spectra for D95V and N97I (EF-hand III variants) showed high degrees of homology with each other, while D131H (EF-hand IV variant) was distinct. Chemical shift perturbation analysis showed that for D95V and N97I, the structure of CaM was perturbed locally, near the site of the mutation (Fig. 6*B*). Due to difficulties in transferring assignments from CaM-WT to the LQTS variants, peak shift analysis could not be performed for many residues (Fig. 6*B*, unassigned residues shown as negative values). For D131H, due to extensive structural changes spanning beyond the region immediate to the site of mutation, the majority of resonances within the C-lobe could not be transferred from the wild-type ($\sim$7% of residues assigned).

### LQTS-associated variants increase CaM susceptibility to protease digestion while temperature sensitivity remains unchanged

To determine the effect of the mutations on the 3D structure of CaM, we investigated the variants' susceptibility to protease digestion. CaM proteins were incubated with various concentrations of trypsin, a serine protease which cleaves peptide chains at the carboxyl-side of lysine and arginine residues. In the absence of Ca$^{2+}$, CaM variants D95V and N97I displayed an increased susceptibility to trypsin digestion, whereas D131H showed a decreased susceptibility, when compared with CaM-WT (Fig. 7*A*, left panel). In the presence of Ca$^{2+}$, CaM proteins required higher concentrations of trypsin to reach near-complete proteolysis, with all mutants showing an increased susceptibility to trypsin proteolysis, when compared with CaM-WT (Fig. 7*A*, right panel).

Using circular dichroism, we monitored the unfolding of apo-CaM as a function of temperature (at 222 nm,

characteristic of $\alpha$-helices). Melting curves were fitted to the Boltzmann sigmoidal equation to determine the midpoint of the unfolding transition (melting temperature, $T_m$) (Fig. 7B, left panel). From the melting curve, we determined the $T_m$ for CaM-WT as $42 \pm 1°C$ ($n = 3$) and no significant differences were observed for LQTS-associated CaM variants. The $T_m$ was $43 \pm 1°C$ for D95V ($n = 3$, $P = 0.0899$), $42 \pm 1°C$ for N97I ($n = 3$, $P = 0.3446$) and $41 \pm 1°C$ for D131H ($n = 3$, $P = 0.1290$).

### CaM interacts with the helix A domain of Kv7.1 in a Ca²⁺-dependent manner

Helix A is a CaM binding domain within the C-terminus of KCNQ1 (Kv7.1-HA$_{370-389}$) (Yus-Najera et al., 2002). Using ITC, we showed that in the absence of $Ca^{2+}$, no measurable interaction between CaM-WT and Kv7.1-HA$_{370-389}$ was observed (Fig. 8A).

In the presence of $Ca^{2+}$, CaM-WT interacted with Kv7.1-HA$_{370-389}$ with an estimated $K_d$ of $96 \pm 10$ $\mu M$ ($n = 3$) (Fig. 8B). The Gibbs free energy change ($\Delta G$) for the interaction between $Ca^{2+}$/CaM and Kv7.1-HA$_{370-389}$ was negative, characteristic of a spontaneous favourable reaction, exothermic and enthalpy-driven.

### LQTS-associated mutations impair interaction of CaM with the helix B domain of Kv7.1

Helix B is a CaM-binding domain within the C-terminus of KCNQ1 (Kv7.1-HB$_{507-536}$) (Yus-Najera et al., 2002). In the absence of $Ca^{2+}$, all CaM variants were found to interact with Kv7.1-HB$_{507-536}$ (Fig. 9, Table 3). The interaction between CaM variants and Kv7.1-HB$_{507-536}$ showed an average stoichiometry of $0.8 \pm 0.1$ across all variants (Fig. 9A). The affinity was significantly decreased for D95V ($K_d = 3.9 \pm 0.1$ $\mu M$) and D131H ($K_d = 7.6 \pm 0.2$ $\mu M$), when compared with CaM-WT ($K_d = 2.1$

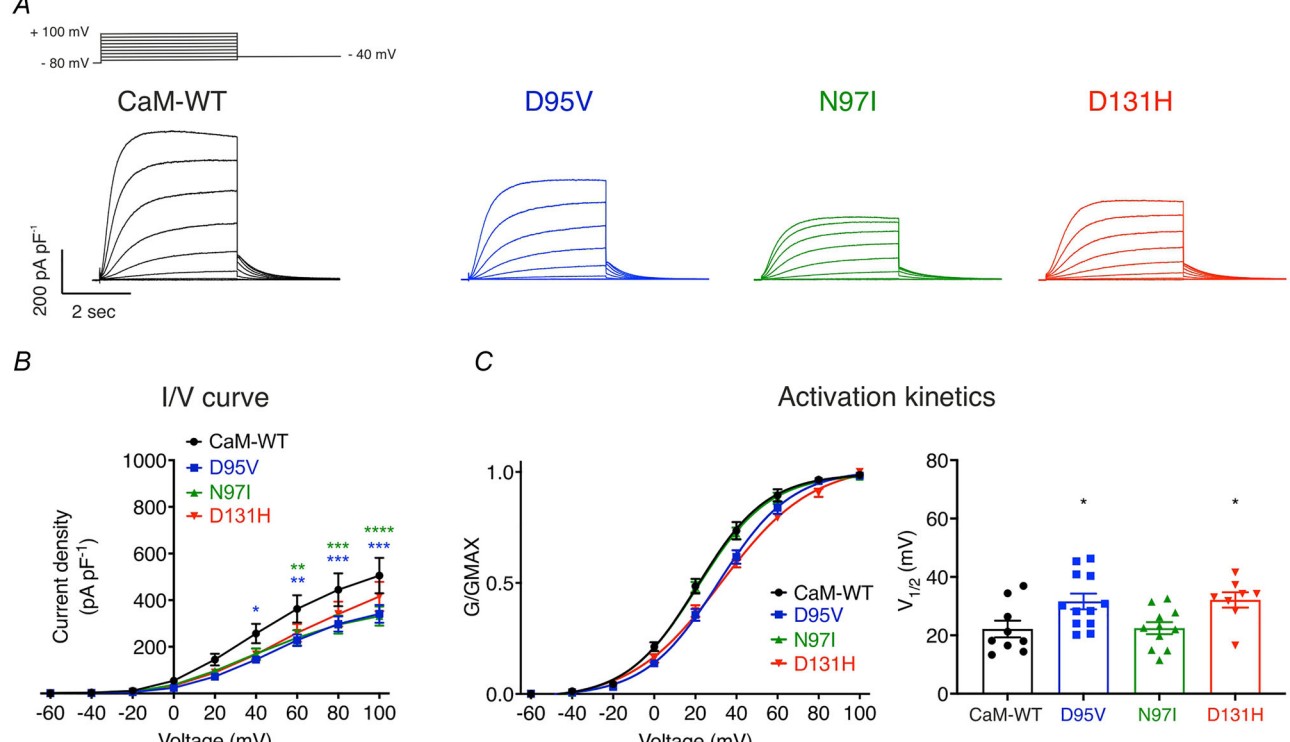

**Figure 3. LQTS-associated CaM mutations decrease IKs densities and reduce voltage sensitivity at high intracellular [Ca²⁺] (1 $\mu M$)**

*A*, representative traces from HEK293T cells transiently transfected with KCNQ1, KCNE1 and CaM variants. Currents were obtained in whole-cell voltage-clamp configuration by holding cells at −80 mV and stepping for 4 s from −60 mV to +100 mV in 20 mV increments, followed by a repolarising step at −40 mV. *B*, current–voltage (I/V) relationships of IKs currents modulated by CaM. Differences between groups were determined using a two-way ANOVA with Dunnett's multiple comparisons tests. *C*, activation kinetics. (Left panel) mean ± s.e.m. Channel conductance, G, normalised to peak conductance, Gmax, to give mean activation/activation curves. (Right panel) mean ± s.e.m. Half maximal activation voltages, V$_{1/2}$, calculated from individual curves fitted using the Boltzmann equation. Differences between groups were determined using a one-way ANOVA with Dunnett's multiple comparisons tests.

± 0.2 μM) (Fig. 9*B*). Binding of all CaM proteins to Kv7.1-HB$_{507-536}$ was energetically favourable (ΔG), endothermic (ΔH), and entropy-driven (ΔS) (Fig. 9*C*). The Gibbs free energy (ΔG) changes were significantly less favourable for LQTS-associated variant D95V (−7.4 ± 0.1 kcal/mol) and D131H (−7.0 ± 0.1 kcal/mol), when compared with CaM-WT (−7.7 ± 0.1 kcal/mol). All mutations conferred significantly more favourable enthalpy changes with reduction in ΔH values from 7.6 ± 0.1 kcal/mol (CaM-WT) to 3.3 ± 0.1 kcal/mol (D131H). All mutations conferred significantly less favourable entropic changes with -TΔS reduced from −15.4 ± 0.1 kcal/mol (CaM-WT) to 10.3 ± 0.1 kcal/mol (D131H).

At saturating Ca$^{2+}$ concentrations, all CaM proteins showed two distinct binding events, at one and two molar excess of Kv7.1-HB$_{507-536}$ to CaM (Fig. 10, Table 4). We observed a tight first interaction ($K_{d1} = 0.56 \pm 0.03$ nM for CaM-WT) followed by a lower affinity second interaction ($K_{d2} = 538 \pm 30$ nM for CaM-WT) (Fig. 10*A*, *B*). All LQTS-associated mutations were found to significantly reduce affinities of both first and second interactions. The affinity was reduced up to 17-fold ($K_{d1} = 9.70 \pm 0.8$ nM for D131H) and 1.7-fold ($K_{d2} = 908 \pm 29$ nM for D131H), for the first and second binding events, respectively, when compared with CaM-WT (Fig. 10*B*, *D*). Thermodynamic signatures revealed that interactions between Ca$^{2+}$/CaM and Kv7.1-HB$_{507-536}$ are energetically

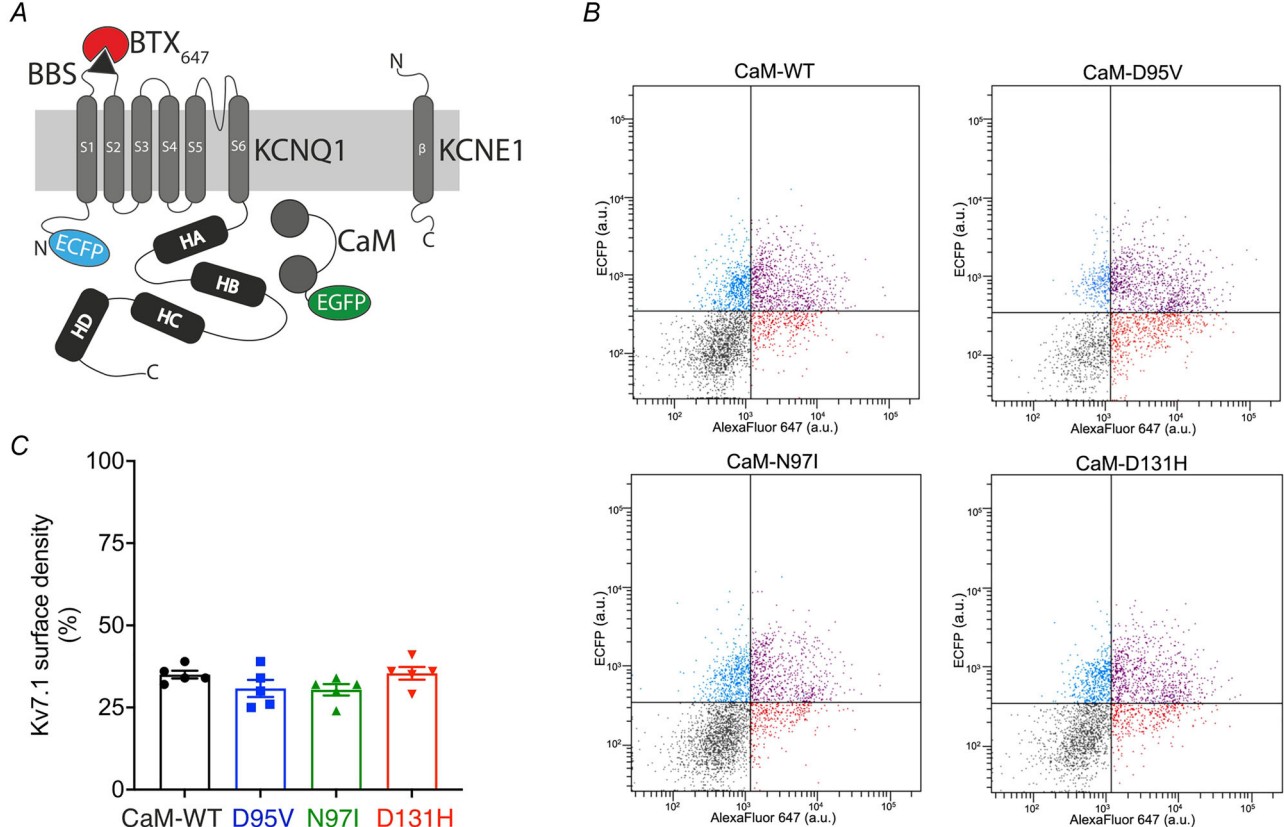

**Figure 4. Optical detection of cell surface density of Kv7.1 with flow cytometry**
*A*, schematic showing Alexa Fluor 647 labelling of cell surface BBS-tagged ECFP-KCNQ1, KCNE1 and CaM-EGFP. HEK293T cells were co-transfected with ECFP-BBS-KCNQ1, KCNE1 and EGFP-labelled CaM at the C-terminus. A bungarotoxin-binding site (BBS) was introduced within the extracellular S1–S2 loop of KCNQ1, allowing for surface labelling using Alexa Fluor 647 conjugated to α-bungarotoxin (BTX$_{647}$). *B*, dot plots of surface-labelled Kv7.1 channels in HEK293T cells expressing ECFP-BBS-KCNQ1, KCNE1 and CaM variants, and incubated with α-bungarotoxin-Alexa Fluor 647. Plots show ECFP *vs.* Alexa Fluor 647 fluorescence (arbitrary units). Ten thousand cells were counted for each experiment. Dots represent a live, single cell as determined after SSC-A/FSC-A and FSC-H/FSC-A gating. Vertical and horizontal lines represent threshold values set based on isochronal and untransfected cells. Top left quadrant (blue) denotes ECFP-BBS-KCNQ1-expressing cells with little Alexa647 signal, indicating low channel surface density. Top right quadrant (purple) represents ECFP-BBS-KCNQ1-positive cells with robust channel trafficking to the surface. Bottom quadrants indicate untransfected cells. *C*, analysis of flow cytometry data to determine the relative surface density for KCNQ1 in the presence of CaM variants (data has been filtered for EGFP-positive cells). Data are expressed as means ± s.e.m. Differences between groups were determined using a one-way ANOVA with Dunnett's multiple comparisons tests.

favourable, exothermic and enthalpy-driven (Fig. 10*C*,*E*). For the first binding event, all LQTS-associated variants showed a reduced ΔG and ΔH as well as altered ΔS, suggesting less energetically favourable interactions and unfavourable conformational changes. For the second binding event, ΔG was reduced for all variants, while ΔH and ΔS were significantly altered only for the N97I and D131H variants, when compared with CaM-WT (Table 4).

### LQTS-associated mutations induce changes in CaM:Kv7.1-HB$_{507-536}$ structure, particularly at saturating Ca$^{2+}$ concentrations

In the absence of Ca$^{2+}$, $^1$H-$^{15}$N HSQC NMR spectral overlays revealed good general consistency between apo-CaM:Kv7.1-HB$_{507-536}$ complexes (Fig. 11*A*). EF-hand III mutant spectra (D95V and N97I) showed near-complete consistency with that of CaM-WT, in contrast to EF-hand IV mutant D131H.

At saturating Ca$^{2+}$ concentrations (1 mM CaCl$_2$), $^1$H-$^{15}$N HSQC NMR spectral overlays showed incomplete homology with the distribution of CaM-WT signals, indicating that LQTS-associated variants proteins adopt alternative conformations when bound to Kv7.1-HB$_{507-536}$ (Fig. 11*B*).

### Predicted effect of IKs inhibition by CaM variants on action potential duration (APD)

First, based on our experimental data, we mathematically altered IKs to match the effect of the LQTS-associated CaM variants (Fig. 1*A*). Then, using the adapted version of the O'Hara–Rudy model, we simulated ventricular action potentials and measured the duration at 50% of the amplitude (APD50). APD50 increased from 216.8 ms (CaM-WT) to 224.7 ms (N97I), 226.7 ms (D131H) and 232.5 ms (D95V). This represents a relative increase of APD of 3.6% (N97I), 4.6% (D131H) and 7.2% (D95V), when compared with CaM-WT (Fig. 12*B-D*).

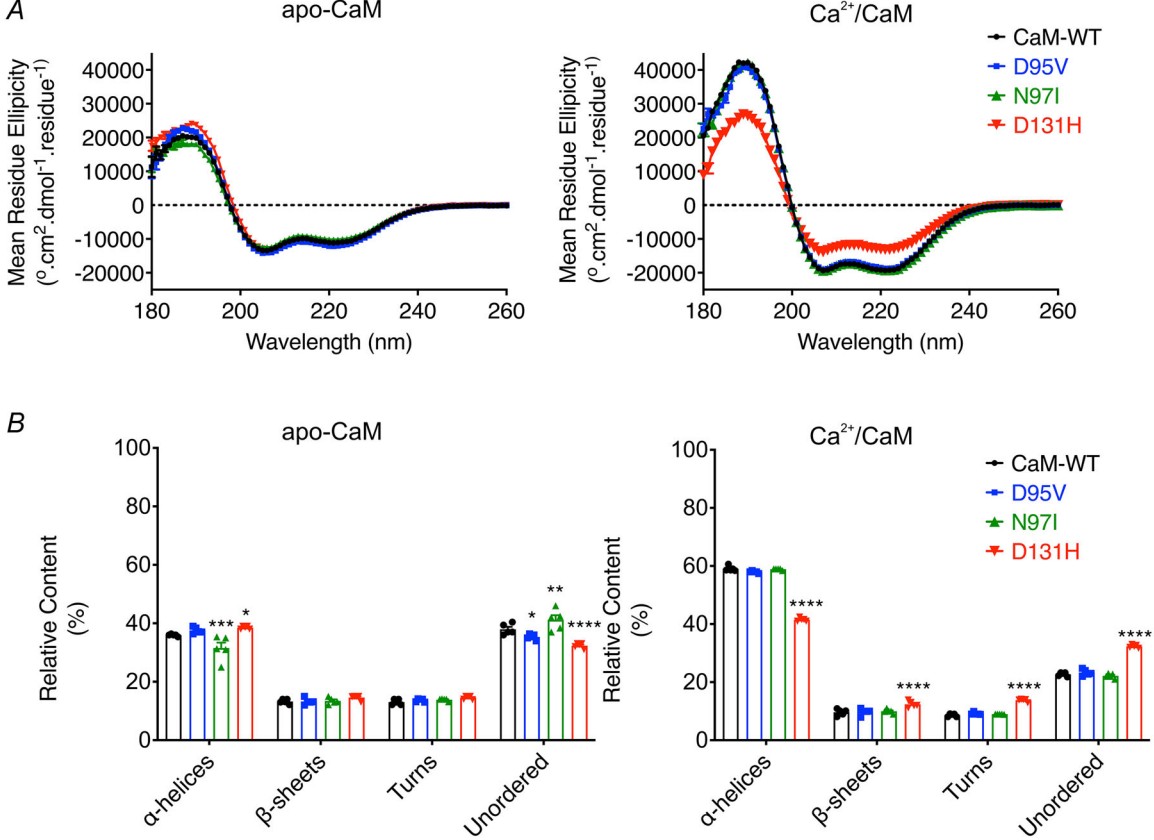

**Figure 5. Arrhythmogenic mutations alter the secondary structure content of CaM**

*A*, circular dichroism spectra of CaM proteins in either 1 mM EGTA (left panel) or 5 mM CaCl$_2$ (right panel). Spectra were collected between 180 and 260 nm at 20°C and buffer-subtracted. *B*, structural distributions predicted using the CDSSTR program (Dichroweb, reference dataset 7) in 1 mM EGTA (left panel) or 5 mM CaCl$_2$ (right panel). Data represent averages of five replicates ± s.e.m. Differences between groups were determined using a two-way ANOVA with Dunnett's multiple comparisons tests.

## Discussion

CaM is a highly conserved, $Ca^{2+}$-sensing protein which has emerged as a modulator of many key proteins and ion channels which govern both excitation–contraction coupling and the spatio-temporal topology of the ventricular action potential. CaM mutations which perturb modulation of these targets promote LQTS (Chazin & Johnson, 2020; Hussey et al., 2023; Jensen et al., 2018), the most common genetic aetiology of which arises from loss-of-function mutations in Kv7.1 (∼50% of cases) (Crotti et al., 2008; Schwartz et al., 2012). CaM facilitates the $Ca^{2+}$-dependent potentiation of IKs (Bai et al., 2005; Bartos et al., 2017; Nitta et al., 1994; Shamgar et al., 2006; Tobelaim et al., 2017; Tohse, 1990) and contributes to proper channel folding, tetramer assembly,

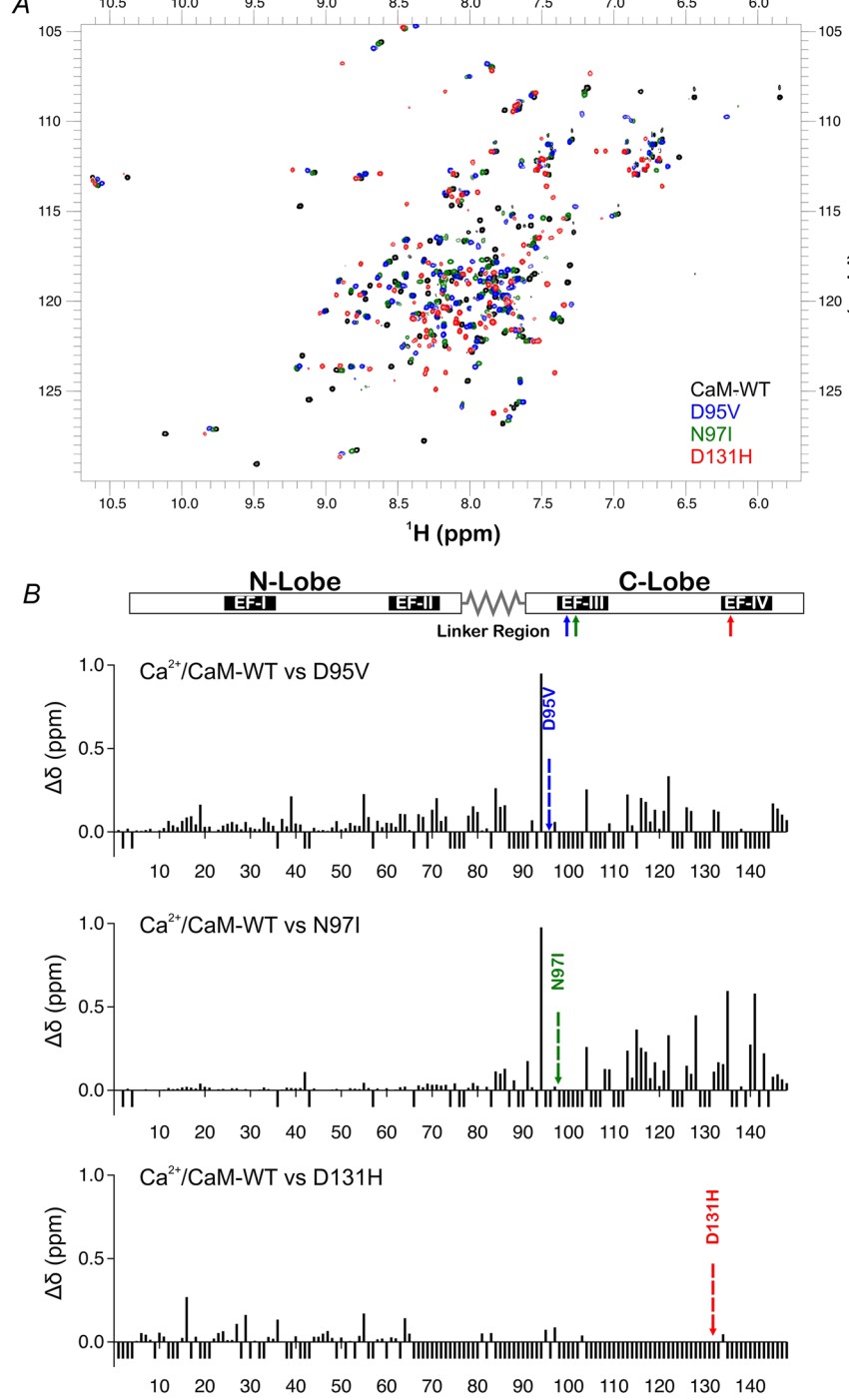

**Figure 6. LQTS-associated mutations induce localised changes in the 3D structure of CaM**

*A*, two dimensional $^{1}$H, $^{15}$N HSQC NMR spectra of $Ca^{2+}$/CaM variants. Overlay of spectra collected from uniformly labelled $^{15}$N CaM proteins in the presence of 1 mM $CaCl_2$. Spectra were collected at 30°C on 700/800 MHz NMR spectrometers (Bruker). *B*, chemical shift perturbation of $Ca^{2+}$-saturated, LQTS-associated CaM mutants compared with CaM-WT. *B*, top panel, schematic of the distribution of key structural features of CaM (N-lobe: 1−72, linker region: 73−87, C-lobe: 88−148). The regions containing the $Ca^{2+}$-binding EF-hands are outlined (EF-hand I: 21−32, EF-hand II: 57−68, EF-hand III: 94−105, EF-hand IV: 130−141). *B*, bottom panels, chemical shift differences ($^{15}$N and $^{1}$H) between the residues of $Ca^{2+}$-saturated CaM-WT and LQTS-associated variants in the presence of 1 mM $CaCl_2$. Residues for which chemical shift differences could not be calculated are shown with an arbitrary value of −0.1 ppm. Chemical shift differences were expressed in ppm as $\Delta\delta = [(\Delta H)^2 + (0.15\Delta N)^2]^{1/2}$.

membrane trafficking and post-translational modification of Kv7.1 (Asada et al., 2009; Ghosh et al., 2006; Shamgar et al., 2006). Disruption of this calmodulation appears to be a prominent driver of arrhythmia, as observed through LQTS-associated Kv7.1 mutations which disrupt CaM interactions at the channel C-terminus (Ghosh et al., 2006; Gonzalez-Garrido et al., 2021; Mousavi Nik et al., 2015; Sachyani et al., 2014; Schmitt et al., 2007; Shamgar et al., 2006; Tobelaim et al., 2017; Yang et al., 2009; Zhou et al., 2016). The consequences of CaM mutants on Kv7.1 function, however, remain elusive. Through adopting a multidisciplinary approach to better understand three LQTS-associated CaM mutants (D95V, N97I

and D131H), we reveal the molecular mechanisms which explain the perturbed structure–function relationships of these CaM variants, how their altered structures hinder complex formation with Kv7.1, and how this modulation results in a LQTS-compatible IKs.

While the modulatory consequences of arrythmia-associated CaM variants have been well characterised in other key cardiac targets such as Cav1.2 and RyR2 (Chazin & Johnson, 2020; Hussey et al., 2023; Jensen et al., 2018), little is known concerning their effects on Kv7.1 (Kato et al., 2022; Rocchetti et al., 2017). Using a HEK293T cellular model, we demonstrated that, at elevated $[Ca^{2+}]_{cyt}$, CaM-WT hastened activation kinetics

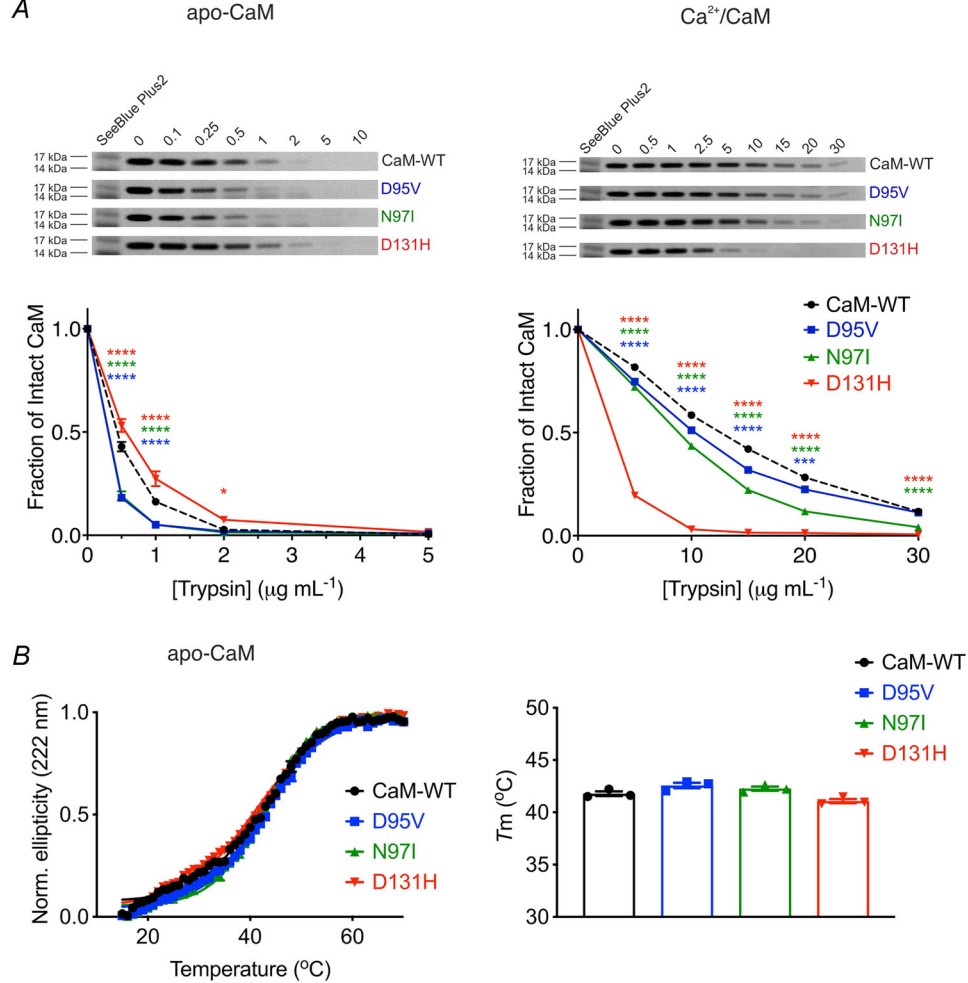

**Figure 7. LQTS-associated CaM mutants show increased susceptibility to protease digestion**
*A*, limited proteolysis of CaM variants in the presence of either (left) 10 mM EGTA or (right) 5 mM CaCl₂. Purified CaM proteins were mixed with increasing concentrations of trypsin for 30 min at 37°C. The fraction of intact CaM was determined by SDS-PAGE and Coomassie staining. Bands were quantified by densitometry analysis using Fiji. *B*, temperature induced unfolding of CaM proteins monitored via circular dichroism at 222 nm in the presence of 1 mM EGTA. *B*, left panel, coloured lines represent averages of triplicate data subject to the Boltzmann sigmoid equation. *B*, right panel, melting points ($T_m$) of CaM proteins derived calculated from interpolating sigmoidal unfolding curves at half-maximal response. Data were normalised and expressed as means ± s.e.m (for each CaM variant, experimental replicates are $n = 4$ in EGTA and $n = 3$ in CaCl₂). Differences between groups were determined using a one-way ANOVA with Dunnett's multiple comparisons tests.

of Kv7.1/KCNE1 and facilitated channel opening at lower membrane potentials when compared with resting $Ca^{2+}$ concentrations (100 nм). These findings agree with previously published work whereby CaM was shown to increase IKs current amplitude and left-shift the voltage dependence of activation in a $Ca^{2+}$-dependent manner (Adam et al., 1993; Nitta et al., 1994; Tobelaim et al., 2017). Such regulation of IKs by CaM is vital in coordinating an appropriate repolarising response to depolarising stimuli (increased $[Ca^{2+}]_{cyt}$) and prevents arrhythmogenic prolongation of the ventricular APD. With regards to the modulatory effects of LQTS-associated CaM variants, we revealed that IKs current density was reduced across a range of membrane potentials at resting $[Ca^{2+}]_{cyt}$ when modulated by all CaM variants, whereas only EF-hand III mutants (D95V and N97I) significantly reduced IKs generated at high $[Ca^{2+}]_{cyt}$. Interestingly, only D95V and

D131H CaM reduced voltage sensitivity of activation at both high and low $[Ca^{2+}]_{cyt}$, suggesting perturbation of the voltage-sensitive domain of Kv7.1. $[Ca^{2+}]_{free}$ of 100 nм and 1 $\mu$м were chosen to mimic the cytosolic environment of a myocyte during diastole and systole, respectively.

Because CaM has been described to regulate trafficking of Kv7.1 to the plasma membrane (Ghosh et al., 2006; Shamgar et al., 2006), the reduced current densities for CaM variants could be associated with a trafficking defect. However, we did not observe any significant impairment of Kv7.1 trafficking to the plasma membrane for any of the LQTS-associated CaM variants assayed, consistent with other studies of mutant CaM-Kv7.1 trafficking (Kato et al., 2022). These data suggest diverse mechanisms by which CaM variants modulate IKs, with reduced $Ca^{2+}$-sensitivity, reduced current amplitude and

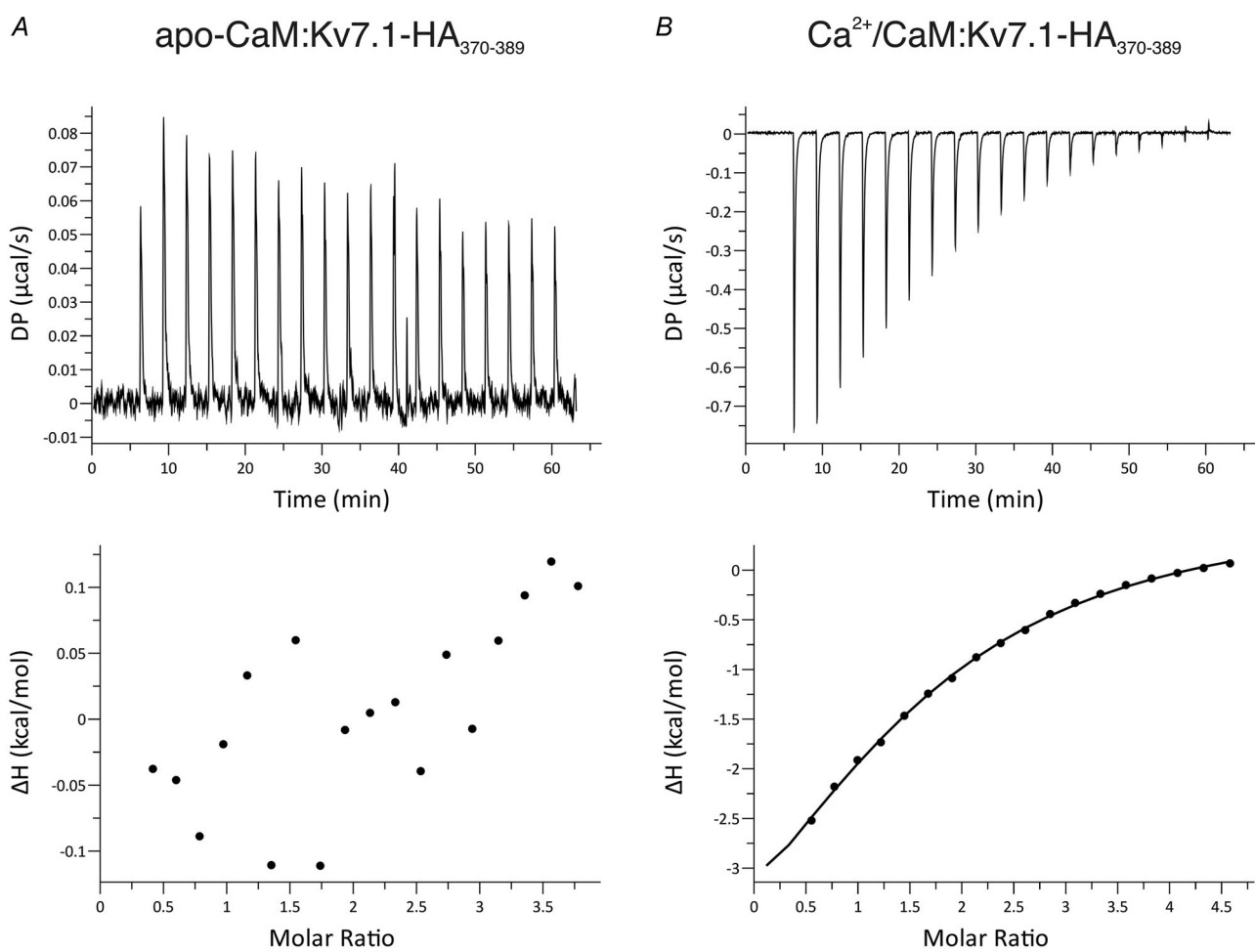

**Figure 8. Interaction between CaM and Kv7.1-HA$_{370-389}$ is $Ca^{2+}$-dependent**
*A, B,* representative isothermal titration calorimetry (ITC) titration curves (upper panel) and binding isotherms (lower panel) for the interaction between CaM and helix A of the Kv7.1 C-terminus (Kv7.1-HA$_{370-389}$) in the presence of (*A*) 1 mм EGTA or (*B*) 5 mм CaCl$_2$. Data were processed using the MicroCal PEAQ-ITC software using a one-site binding model. The sum of the change in enthalpy ($\Delta$H) and the change in entropy ($\Delta$S) multiplied by the absolute temperature (T) gives the change in free energy ($\Delta$G). Experiments were performed at 25°C. DP, differential power.

depolarised voltage dependence of activation (positive shift in $V_{1/2}$) observed, all of which contribute to a LQTS-compatible IKs current.

To decipher the mechanisms behind this pathogenic modulation, the conformational consequences of structural perturbations to CaM were investigated. Circular dichroism experiments revealed that LQTS-CaM variants were less able to undergo $Ca^{2+}$-induced conformational change, a structural transition essential to CaM's function as a $Ca^{2+}$-signalling protein. This was most apparent in EF-hand IV variant D131H and agrees with other studies where differences in structural distributions for CaM-mutant compared with CaM-WT were observed (Dal Cortivo et al., 2022; Hennessey et al., 1987). Using chemical shift analysis of HSQC NMR spectra to more precisely identify areas of structural change, we revealed that in their $Ca^{2+}$-saturated states, D95V and N97I presented

with localised perturbation within the C-lobe, while perturbations to D131H were global, affecting both the N-lobe and the C-lobe. These findings agree with other studies which have demonstrated a site-dependent, ranging degree of chemical shift perturbation across LQTS-CaM variants, which present more apparently when variants are compared with WT in their calcified states (Holt et al., 2020; Pipilas et al., 2016; Wang et al., 2020; Wren et al., 2019). Additionally, we demonstrated that LQTS-associated CaM variants displayed altered susceptibility to protease hydrolysis, consistent with other works (Crotti et al., 2013; Dal Cortivo et al., 2022; Prakash et al., 2023). This was more apparent when mutants were compared in $Ca^{2+}$-saturating conditions, reflecting the more distinct $Ca^{2+}$-bound conformations of mutants, rather than their more subtly dissimilar apo-conformations. A specific order of chelation exists within the four EF-hands of CaM. Precise allosteric

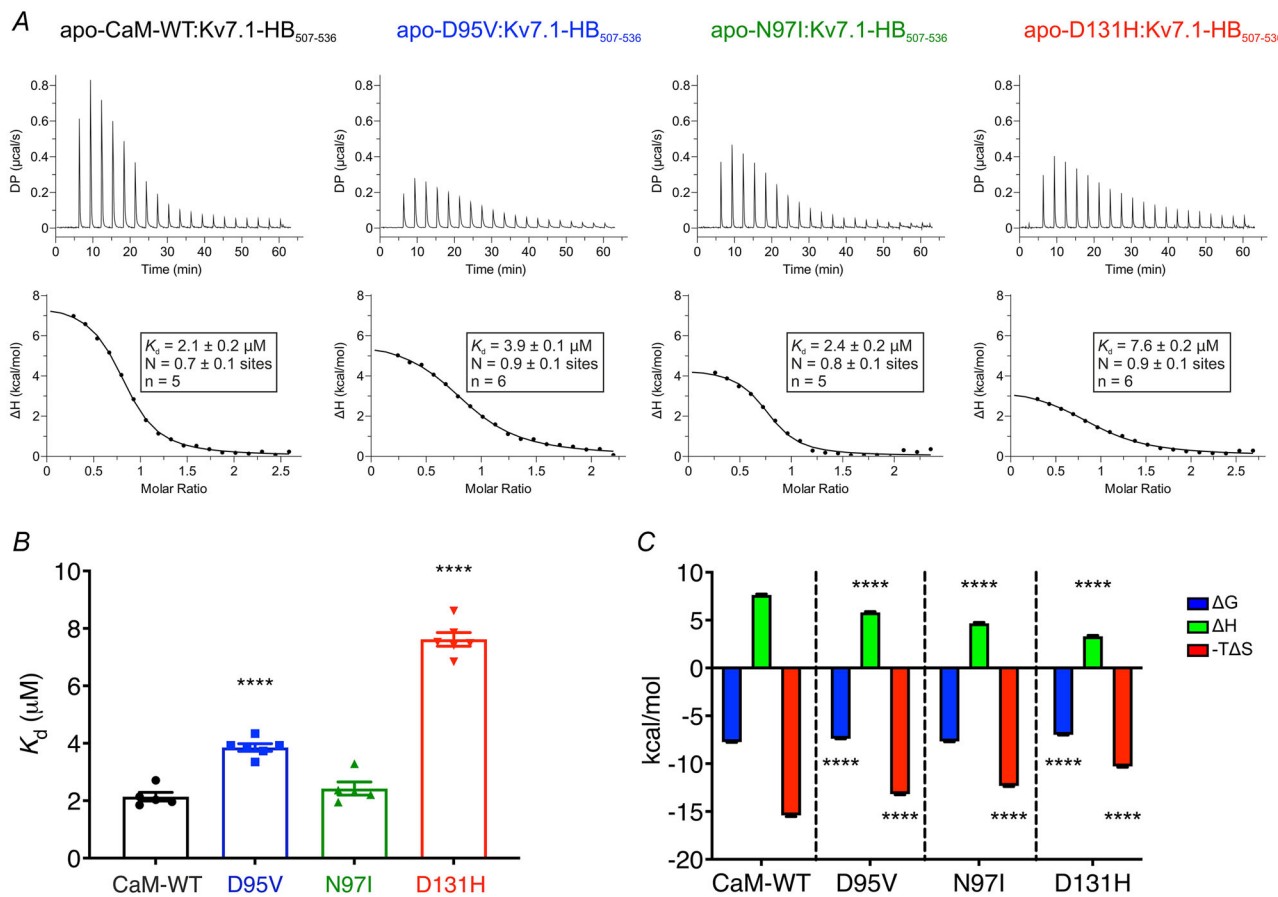

**Figure 9. Apo-CaM binding to Kv7.1-HB$_{507-536}$ is decreased for LQTS-associated variants**
*A*, representative ITC titration curves (upper panel) and binding isotherms (lower panel) for the interaction between apo-CaM and helix B of the Kv7.1 C-terminus (Kv7.1-HB$_{507-536}$). *B*, affinity and (*C*) thermodynamic profile of the binding of apo-CaM to Kv7.1-HB$_{507-536}$ obtained by fitting to a one-site binding model. Data are means ± s.e.m. N, stoichiometry; n, number of experimental replicates. The sum of the change in enthalpy (ΔH) and the change in entropy (ΔS) multiplied by the absolute temperature (T) gives the change in free energy (ΔG). Experiments were performed at 25°C in the presence of 1 mM EGTA. DP, differential power. Differences between groups were determined using a one-way ANOVA with Dunnett's multiple comparisons tests.

**Table 3. Summary of the binding constants and thermodynamic parameters for the interaction between apo-CaM variants and Kv7.1 (Helix B). Stoichiometry (N), dissociation constant ($K_d$), enthalpy change ($\Delta H$), entropy change ($-T\Delta S$) and Gibbs free energy ($\Delta G$) were obtained from fitting the data to a one-site binding model. Values are provided as means ± s.e.m. Statistical significance was determined using one-way ANOVA with Dunnett's multiple comparisons tests.**

| Variant | N (sites) | $K_d$ (μM) | P value (number of replicates) | $\Delta H$ (kcal/mol) | P value (number of replicates) | $-T\Delta S$ (kcal/mol) | P value (number of replicates) | $\Delta G$ (kcal/mol) | P value (number of replicates) |
|---|---|---|---|---|---|---|---|---|---|
| CaM-WT | 0.7 ± 0.1 | 2.1 ± 0.2 | (5) | 7.6 ± 0.1 | (5) | −15.4 ± 0.1 | (5) | −7.7 ± 0.1 | (5) |
| D95V | 0.9 ± 0.1 | 3.9 ± 0.1 | <0.0001 (6) | 5.8 ± 0.1 | <0.0001 (6) | −13.2 ± 0.1 | <0.0001 (6) | −7.4 ± 0.1 | <0.0001 (6) |
| N97I | 0.8 ± 0.1 | 2.4 ± 0.2 | 0.6303 (5) | 4.7 ± 0.1 | <0.0001 (5) | −12.3 ± 0.1 | <0.0001 (5) | −7.7 ± 0.1 | 0.3112 (5) |
| D131H | 0.9 ± 0.1 | 7.6 ± 0.2 | <0.0001 (6) | 3.3 ± 0.1 | <0.0001 (6) | −10.3 ± 0.1 | <0.0001 (6) | −7.0 ± 0.1 | <0.0001 (6) |

regulation and positive cooperativity between $Ca^{2+}$ binding sites facilitate a system whereby EF-hand IV binds $Ca^{2+}$ first, then EF-hand III, followed by EF-hand II and finally EF-hand I (Liu et al., 2019). Therefore, mutations which reduce the $Ca^{2+}$ binding of EF-hand IV are likely to disrupt downstream calcification of all other EF-hands, whereas disruptions to EF-hand III would still benefit from the positive cooperativity from calcification of EF-hand IV. Mutations within EF-hand IV typically reduce $Ca^{2+}$ binding affinity more so than mutations in EF-hand III (Jensen et al., 2018) and so help explain how D131H appears to more significantly perturb the structure of CaM compared with D95V and N97I. The data outline the significant effects which single missense, C-lobe mutations infer on the conformation of CaM.

To better understand how LQTS-CaM mutants infer altered modulation of IKs, the interactions between CaM proteins and their isolated binding domains of Kv7.1 were characterised. We showed that CaM interacts with Kv7.1-HA$_{370-389}$ exclusively in the presence of $Ca^{2+}$, similar to works performed on Kv7.4 (Archer et al., 2019). The affinity of this interaction was low and could not be accurately determined for the LQTS-associated CaM variants. While the apo-CaM:Kv7.1-HA$_{370-389}$ could not be fully characterised, crystal structures of the CaM-KCNQ1 complex at HA and HB consistently reveal the apo C-lobe of CaM bound to HA, even in high molar excesses of $Ca^{2+}$ (Sachyani et al., 2014). This has also been suggested in pull down assays between the C-terminus of KCNQ1 and CaM (Tobelaim et al., 2017). For Kv7.1-HB$_{507-536}$, we showed that CaM can interact in both $Ca^{2+}$-independent and dependent manners. In apo conditions, HSQC NMR spectral overlays revealed good general consistency between apo-CaM:Kv7.1-HB$_{507-536}$ complexes. EF-hand III mutant spectra (D95V and N97I) showed near-complete overlap with CaM-WT, contrasting with EF-hand IV mutant D131H and further supporting the more apparent conformational divergence of the EF-hand IV variant. At saturating $Ca^{2+}$ concentrations, the HSQC NMR spectral overlays revealed incomplete homology with the distribution of CaM-WT signals, indicating that LQTS-associated variant proteins adopt alternative conformations when bound to Kv7.1-HB$_{507-536}$. Using ITC, we showed that the apo-CaM:Kv7.1-HB$_{507-536}$ interaction was driven by hydrophobic interactions, with all apo-mutants revealing less favourable conformational changes (reflected by their significantly increased $-T\Delta S$ contributions). Variants D95V and D131H exhibited significant increases in measured $K_d$. In the presence of $Ca^{2+}$, interaction of CaM variants with Kv7.1-HB$_{507-536}$ was driven by hydrogen bonding and Van der Waals forces (reflected in negative $\Delta H$ values). For all LQTS-associated mutants, we observed a reduction in affinity. The trends across data appear consistent, whereby EF-hand IV mutant (D131H)

confers the greatest reduction in affinity for Kv7.1, most notably in $Ca^{2+}$-saturating conditions. Compared with EF-hand III variants (D95V and N97I), this mutant was most structurally distinct from WT-CaM, particularly when compared in the presence of $Ca^{2+}$.

Together, the findings presented here reveal that single residue substitutions to CaM's highly conserved structure have substantial effects on global protein structure. In the case of the variants studied (D95V, N97I and D131H), substitutions occur at residues which

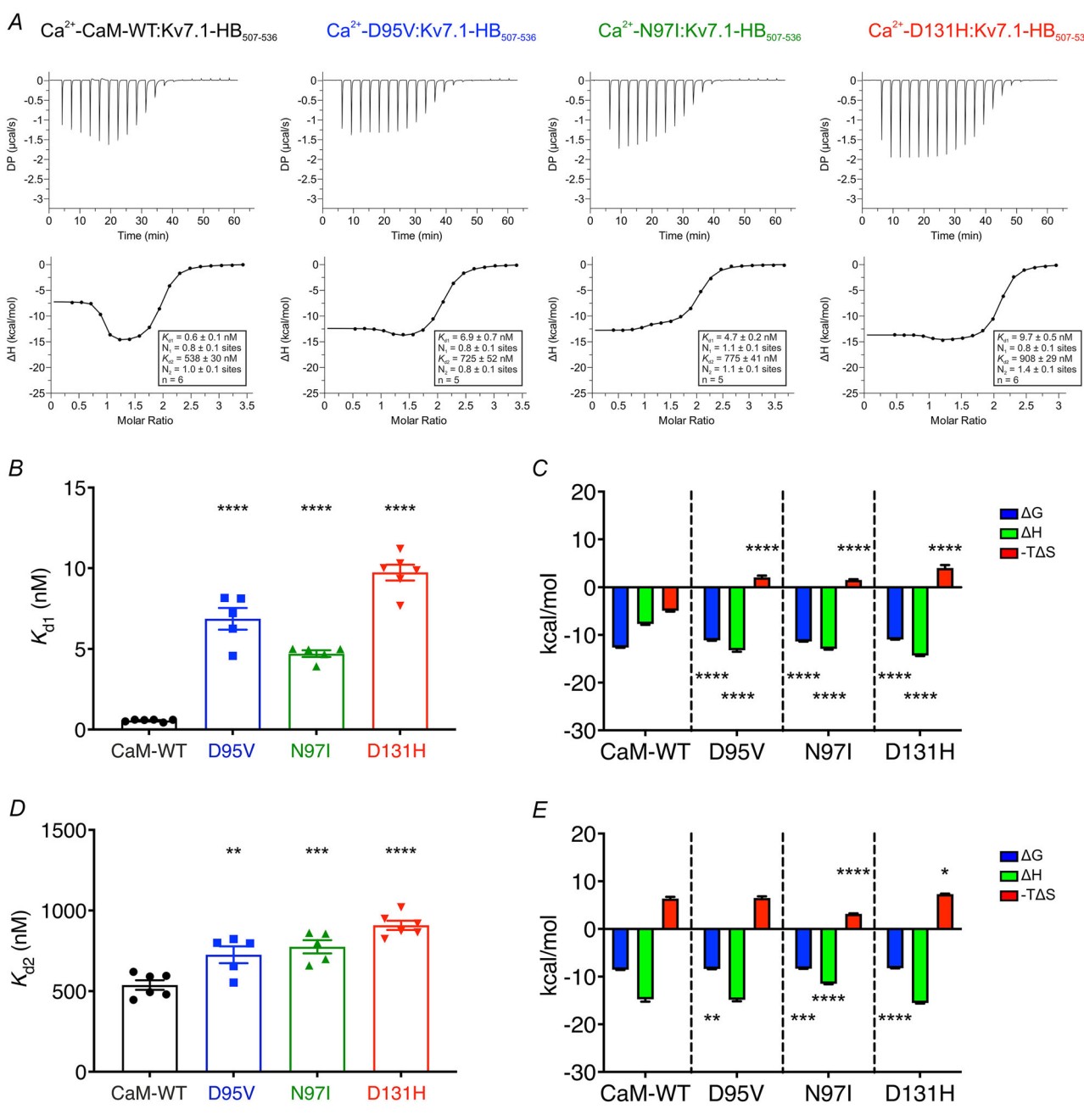

**Figure 10. $Ca^{2+}$-CaM binding to Kv7.1-HB$_{507-536}$ is decreased for LQTS-associated variants**
*A*, representative ITC titration curves (upper panel) and binding isotherms (lower panel) for the interaction between apo-CaM and helix B of the Kv7.1 C-terminus (Kv7.1-HB$_{507-536}$). *B, D*, affinity and (*C, E*) thermodynamic profile of the binding of apo-CaM to Kv7.1-HB$_{507-536}$ obtained by fitting to a two-site binding model. *B, C*, binding parameters for the first interaction and (*D, E*) for the second interaction. Data are means ± s.e.m. N, stoichiometry; n, number of experimental replicates. The sum of the change in enthalpy ($\Delta H$) and the change in entropy ($\Delta S$) multiplied by the absolute temperature (T) gives the change in free energy ($\Delta G$). Experiments were performed at 25°C in the presence of 5 mM $CaCl_2$. DP, differential power. Differences between groups were determined using a one-way ANOVA with Dunnett's multiple comparisons tests.

**Table 4. Summary of the binding constants and thermodynamic parameters for the interaction between Ca²⁺/CaM variants and Kv7.1 (Helix B). Stoichiometry (N), dissociation constant (Kd), enthalpy change (ΔH), entropy change (−TΔS) and Gibbs free energy (ΔG) were obtained from fitting the data to a two-site binding model. Values are provided as means ± s.e.m. Statistical significance was determined using one-way ANOVA with Dunnett's multiple comparisons tests.**

| Variant | N (sites) | $K_d$ (nM) | P value (number of replicates) | ΔH (kcal/mol) | P value (number of replicates) | −TΔS (kcal/mol) | P value (number of replicates) | ΔG (kcal/mol) | P value (number of replicates) |
|---|---|---|---|---|---|---|---|---|---|
| CaM-WT | 0.8 ± 0.1 | 0.6 ± 0.1 | (6) | −7.7 ± 0.1 | (6) | −4.9 ± 0.2 | (6) | −12.7 ± 0.1 | (6) |
|  | 1.0 ± 0.1 | 538 ± 30 | (6) | −14.8 ± 0.5 | (6) | 6.4 ± 0.4 | (6) | −8.6 ± 0.1 | (6) |
| D95V | 0.8 ± 0.1 | 6.9 ± 0.7 | <0.0001 (5) | −13.2 ± 0.3 | <0.0001 (5) | 2.1 ± 0.3 | <0.0001 (5) | −11.1 ± 0.1 | <0.0001 (5) |
|  | 0.8 ± 0.1 | 725 ± 52 | 0.0067 (5) | −14.9 ± 0.3 | 0.9906 (5) | 6.5 ± 0.3 | 0.9749 (5) | −8.4 ± 0.1 | 0.0048 (5) |
| N97I | 1.1 ± 0.1 | 4.7 ± 0.2 | <0.0001 (5) | −12.9 ± 0.2 | <0.0001 (5) | 1.5 ± 0.2 | <0.0001 (5) | −11.4 ± 0.1 | <0.0001 (5) |
|  | 1.1 ± 0.1 | 775 ± 41 | 0.0009 (5) | −11.5 ± 0.1 | <0.0001 (5) | 3.1 ± 0.1 | <0.0001 (5) | −8.3 ± 0.1 | 0.0007 (5) |
| D131H | 0.8 ± 0.1 | 9.7 ± 0.5 | <0.0001 (6) | −14.3 ± 0.1 | <0.0001 (6) | 4.0 ± 0.6 | <0.0001 (6) | −11.0 ± 0.1 | <0.0001 (6) |
|  | 1.4 ± 0.1 | 908 ± 29 | <0.0001 (6) | −15.5 ± 0.1 | 0.2136 (6) | 7.3 ± 0.1 | 0.0464 (6) | −8.2 ± 0.1 | <0.0001 (6) |

directly coordinate Ca²⁺ at the C-lobe of the protein, resulting in reduced Ca²⁺-binding (Crotti et al., 2013; Makita et al., 2014; Pipilas et al., 2016; Sondergaard et al., 2015; Vassilakopoulou et al., 2015) and reduced conformational plasticity in response to increases in [Ca²⁺]. The perturbed structures of CaM mutants were found to reduce their affinity of interaction at the C-terminus of Kv7.1, both in the absence and presence of calcium. Ca²⁺-dependent binding was significantly weakened compared with Ca²⁺-independent interactions of mutants, reflected by their more divergent calcified conformations than apo ones. The conformations adopted by CaM mutants with their binding sites were significantly distinct from those adopted by WT CaM, with exacerbations to perturbation found more so in the presence of Ca²⁺ than in Ca²⁺-free conditions. The alternative structures which CaM mutants adopt with Kv7.1 were found to result in aberrant channel modulation, while trafficking remained unaffected. Ca²⁺-sensitivity of LQTS-CaM modulated Kv7.1 channels were reduced and generated smaller IKs when elicited in conditions which mimicked those of physiological cardiac contraction (1 μM [Ca²⁺]$_{free}$). Such modulation would reduce the repolarisation capacity of IKs, extending the APD and contributing to the LQTS phenotype which presented in patients harbouring said CaM mutations. LQTS-associated CaM variants have been shown to perturb a range of other cardiac ion channels, including reducing the Ca²⁺-dependent inactivation of Cav1.2 (Gomez-Hurtado et al., 2016; Limpitikul et al., 2014; Prakash et al., 2023; Yin et al., 2014), altered inhibition of RyR2 (Nomikos et al., 2014; Vassilakopoulou et al., 2015) and activation of CaMKIIδ (Berchtold et al., 2016; Prakash et al., 2023).

The large repertoire of targets which CaM interacts with, combined with the limited clinical data from CaM-driven LQTS patients, make it difficult to establish a direct correlation between CaM mutation and disease severity. Using mathematical modelling, we predicted that IKs reduction for the LQTS-CaM variants would cause an APD prolongation of 3.6% (N97I), 4.6% (D131H) and 7.2% (D95V). Considering that QTc interval is prolonged by 26% (N97I, from 440 to 555 ms) (Crotti et al., 2013), 48% (D131H, from 440 to 651 ms) (Makita et al., 2014) and 57% (D95V, from 440 to 690 ms) (Pipilas et al., 2016), data suggest a substantial IKs contribution to the LQTS phenotype across all variants: 14% (N97I), 10% (D131H) and 13% (D95V). It should be appreciated that the functional effects of CaM mutants described here would span beyond Kv7.1 modulation and would simultaneously exacerbate multiple modalities of LQTS (enhanced $I_{Ca,L}$, $I_{Na}$). We demonstrate here that LQTS-CaM mutants have the capacity to prolong membrane repolarisation, CaM mutants have also been demonstrated to perturb Cav1.2 inactivation and could modulate a range of

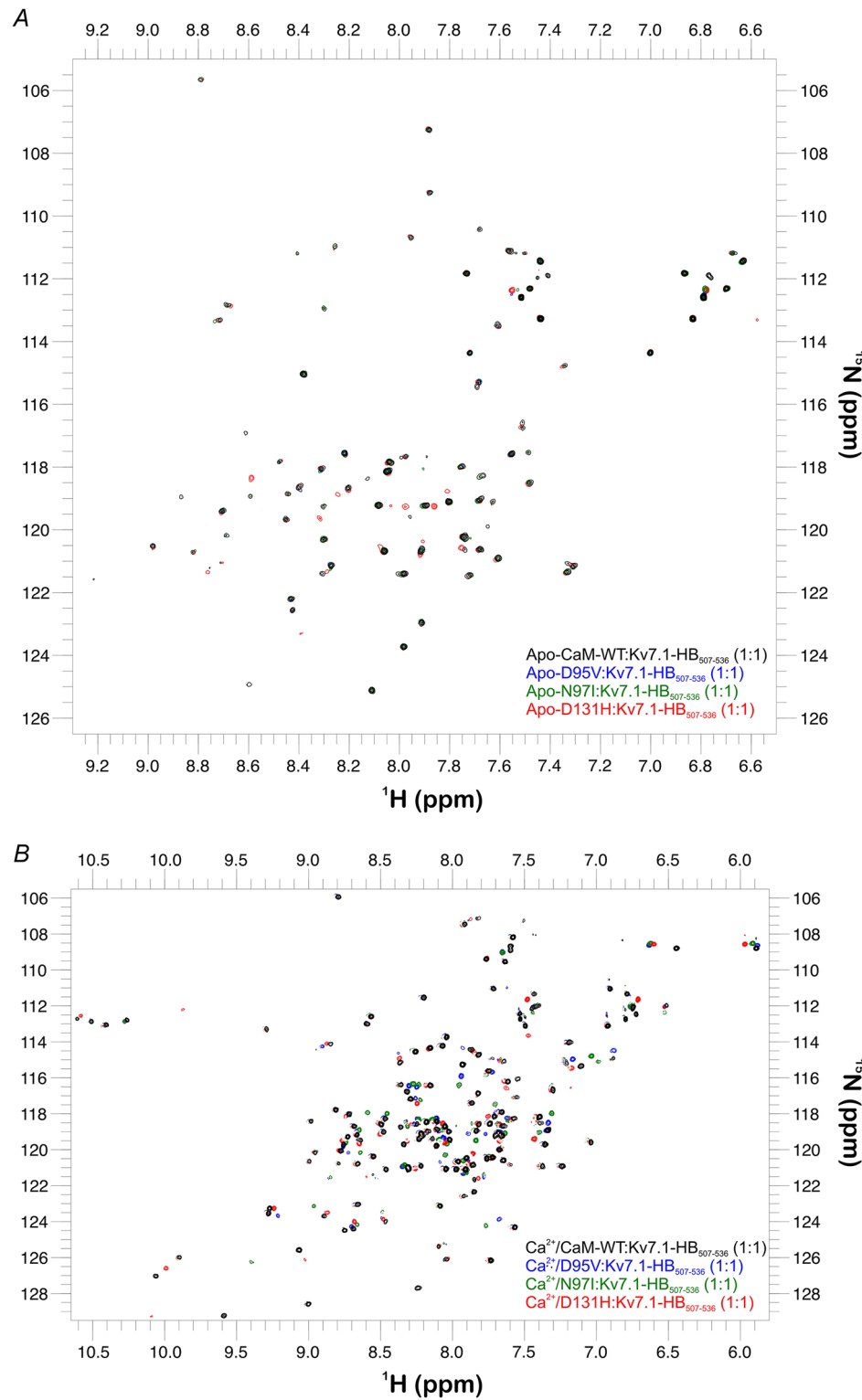

**Figure 11. Arrhythmogenic mutations induce changes in the 3D structure of the CaM:Kv7.1-HB$_{507-536}$ complex**

*A*, two dimensional $^1$H, $^{15}$N HSQC NMR spectra of CaM variants in complex with Kv7.1-HB$_{507-536}$. Overlay of spectra collected from uniformly labelled $^{15}$N CaM proteins in the presence one molar equivalent of Kv7.1-HB$_{507-536}$ and (*A*) 1 mм EGTA or (*B*) 1 mм CaCl$_2$. Spectra were collected at 30°C on 700/800 MHz NMR spectrometers (Bruker).

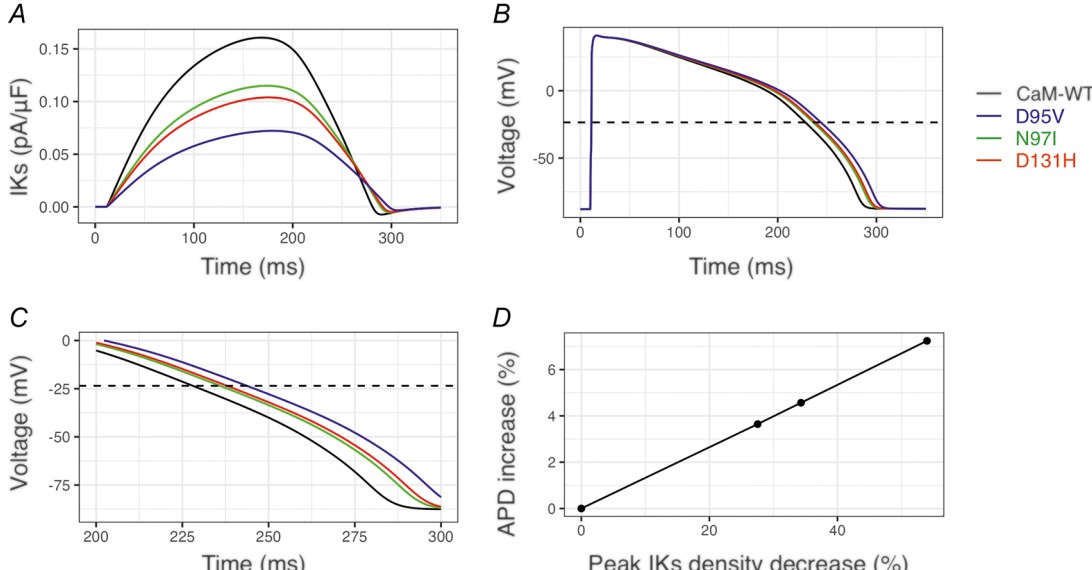

**Figure 12. Predicted effects of direct IKs inhibition by LQTS-associated CaM variants**
Simulations run in CellML using an adapted version of the O'Hara–Rudy model (O'Hara et al., 2011), see Methods. *A*, different values of Kb (eqn 2, see Methods) were used until the resulting block of peak IKs matched that of the experimental data for CaM-WT, D95V, N97I and D131H variants. *B*, output simulated ventricular action potentials for each of these situations. The dotted line indicates the action potential 50% level (APD50) where action potential duration has been measured. *C*, expanded view of the simulated action potentials shown in panel *B*. *D*, plot of APD duration increase for the given reduction in IKs current density. In each of *A,B*, the waveforms are colour coded to the simulation condition: black is CaM-WT, blue is D95V, green is N97I and red is D131H.

arrhythmogenic substrates (Benitah et al., 2010; January et al., 1988; Madhvani et al., 2015; Weiss et al., 2010). The most common mechanism of arrythmia involves the generation of early-after depolarisations (EADs) via self-amplification of $I_{Ca,L}$ (Weiss et al., 2010). EADs promote discordant refractory periods across the myocardium and create functional obstacles through which the depolarising wave of activity must navigate (Sato et al., 2009; Weiss et al., 2010). The results in a multifocal, self-sustaining arrhythmia where the site of origin continuously shifts throughout the tissue, producing the characteristic 'Torsade de Pointes', which can lead to ventricular fibrillation.

In summary, our study demonstrates that LQTS-associated single residue substitutions to CaM significantly disrupt its structure–function relationship as a $Ca^{2+}$-sensing protein. Pathogenic CaM variants; D95V, N97I and D131H contribute to aberrant IKs generation through adopting alternative conformations with Kv7.1. This work aids in elucidating the site-dependent effects of CaM mutations and highlights the significant disease processes which CaM mutants underpin. Pharmacologically, LQTS patients are usually treated with $\beta$-blockers (Farzam & Jan, 2022). While broadly effective, many patients still suffer from recurrent arrhythmic episodes while treated with $\beta$-blockers. The multiple genotypes which contribute to LQTS therefore likely reduce the effectiveness of one blanket pharmacological

agent (Ahn et al., 2017). This results in almost one third (32%) of LQTS patients experiencing recurrent cardiac events despite being on $\beta$-blocker therapies (Moss et al., 2000). Specifically, LQTS patients with CaM mutations respond varyingly to these therapies (Ahn et al., 2017). Understanding the molecular mechanisms by which CaM can aberrantly modulate targets has proven valuable and allows clinicians to better guide management of patients harbouring CaM mutations (Webster et al., 2017).

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

## Additional information

### Data availability statement

The data that support the findings of this study are available from the corresponding author, NH, upon reasonable request.

### Competing interests

The authors declare that they have no known competing financial interests or personal relationships that could have appeared to influence the work reported in this paper.

### Author contributions

L. M., C.D. and N. H. conceptualization; L. M., K.W., A. M., N. G., R. M., M. H., O.P., J. C., R. B-J., C. D. and N. H. methodology; L. M., K. W., A. M., R. M., R.B-J., C.D. and N. H. formal analysis; L. M., K.W., A. M., N.G., R. M., M.H., O. P., J.C., R.B-J., C.D. and N.H. investigation; L.M., K. W., A.M., R. M., R.B-J., C. D. and N. H. writing–original draft; L. M., K. W., A. M., N.G., R. M., M.H., J.C., R. B-J., C.D. and N.H. writing–review and editing; C. D. and N. H. funding acquisition; C. D. and N. H. supervision. All authors have approved the final version of the manuscript and agree to be accountable for all aspects of the work. All persons designated as authors qualify for authorship, and all those who qualify for authorship are listed.

## Funding

This work was supported by British Heart Foundation Inter-mediate Basic Science Research Fellowship (FS/17/56/32 925 and FS/EXT/22/35 014 to N. H.), British Heart Foundation Project Grant (PG/21/10 521 to N. H.), BBSRC grant (BB/V002767/1 to C. D.), British Heart Foundation Non-clinical PhD studentships (FS/PhD/20/29 025 and FS/PhD/22/29 339 to N. H.), Wellcome Trust 4-year PhD studentship programme (102 172/B/13/Z to N. G.) and University of Liverpool, Institute of Translational Medicine PhD studentship (to L. M.).

## Acknowledgements

The authors would like to thank Dr Christopher Law and Dr Sandra Pereira Cachinho from the Cell Sorting and Mass cytometry Facility (University of Liverpool) for their technical support.

## Keywords

calcium, calmodulin, cardiac arrhythmia, IKs, Kv7.1, LQTS

## Supporting information

Additional supporting information can be found online in the Supporting Information section at the end of the HTML view of the article. Supporting information files available:

**Statistical Summary Document**
**Peer Review History**

