## [Peer Review History · The Journal of Physiology]

Long QT syndrome-associated calmodulin variants disrupt the activity of the slowly activating delayed rectifier potassium channel (IKs).

Liam F McCormick, Kirsty Wadmore, Amy Milburn, Nitika Gupta, Rachael Morris, Marie Held, Ohm Prakash, Joseph Carr, Richard Barrett-Jolley, Caroline Dart, and Nordine Helassa
DOI: 10.1113/JP284994

Corresponding author(s): Nordine Helassa (nhelassa@liverpool.ac.uk)

Review Timeline:	Submission Date:	08-May-2023
	Editorial Decision:	24-May-2023
	Revision Received:	15-Jun-2023
	Accepted:	21-Jun-2023

Senior Editor: Natalia Trayanova

Reviewing Editor: Eleonora Grandi

Transaction Report:

Dear Dr Helassa,

Re: JP-RP-2023-284994 "Long QT syndrome-associated calmodulin variants disrupt the activity of the slowly activating delayed rectifier potassium channel (IKs)" by Liam F McCormick, Kirsty Wadmore, Amy Milburn, Nitika Gupta, Rachael Morris, Marie Held, Ohm Praksah, Joseph Carr, Caroline Dart, and Nordine Helassa

Thank you for submitting your manuscript to The Journal of Physiology. It has been assessed by a Reviewing Editor and by 2 expert referees and we are pleased to tell you that it is potentially acceptable for publication following satisfactory major revision.

LANGUAGE EDITING AND SUPPORT FOR PUBLICATION: If you would like help with English language editing, or other article preparation support, Wiley Editing Services offers expert help, including English Language Editing, as well as translation, manuscript formatting, and figure formatting at www.wileyauthors.com/eoo/preparation. You can also find resources for Preparing Your Article for general guidance about writing and preparing your manuscript at www.wileyauthors.com/eoo/prepresources.

REVISION CHECKLIST:

We look forward to receiving your revised submission.

Yours sincerely,

Natalia Trayanova
Senior Editor
The Journal of Physiology

REQUIRED ITEMS FOR REVISION

-Author photo and profile. First (or joint first) authors are asked to provide a short biography (no more than 100 words for one author or 150 words in total for joint first authors) and a portrait photograph. These should be uploaded and clearly labelled with the revised version of the manuscript. See Information for Authors for further details.

-The Reference List must be in Journal format

-Your manuscript must include a complete Additional Information section

-Please upload separate high-quality figure files via the submission form.

-Please ensure that the Article File you upload is a Word file.

-A Statistical Summary Document, summarising the statistics presented in the manuscript, is required upon revision. It must be on the Journal's template, which can be downloaded from the link in the Statistical Summary Document section here: https://jp.msubmit.net/cgi-bin/main.plex?form_type=display_requirements#statistics

-Papers must comply with the Statistics Policy https://jp.msubmit.net/cgi-bin/main.plex?form_type=display_requirements#statistics

In summary:

-If $n \leq 30$, all data points must be plotted in the figure in a way that reveals their range and distribution. A bar graph with data points overlaid, a box and whisker plot or a violin plot (preferably with data points included) are acceptable formats.

-If $n > 30$, then the entire raw dataset must be made available either as supporting information, or hosted on a not-for-profit repository e.g. FigShare, with access details provided in the manuscript.

- n clearly defined (e.g. x cells from y slices in z animals) in the Methods. Authors should be mindful of pseudoreplication.

-All relevant n values must be clearly stated in the main text, figures and tables, and the Statistical Summary Document (required upon revision)

-The most appropriate summary statistic (e.g. mean or median and standard deviation) must be used. Standard Error of the Mean (SEM) alone is not permitted.

-Exact p values must be stated. Authors must not use 'greater than' or 'less than'. Exact p values must be stated to three significant figures even when 'no statistical significance' is claimed.

-Statistics Summary Document completed appropriately upon revision

-Please include an Abstract Figure file, as well as the figure legend text within the main article file. The Abstract Figure is a piece of artwork designed to give readers an immediate understanding of the research and should summarise the main conclusions. If possible, the image should be easily 'readable' from left to right or top to bottom. It should show the physiological relevance of the manuscript so readers can assess the importance and content of its findings. Abstract Figures should not merely recapitulate other figures in the manuscript. Please try to keep the diagram as simple as possible and without superfluous information that may distract from the main conclusion(s). Abstract Figures must be provided by authors no later than the revised manuscript stage and should be uploaded as a separate file during online submission labelled as File Type 'Abstract Figure'. Please ensure that you include the figure legend in the main article file. All Abstract Figures should be created using BioRender. Authors should use The Journal's premium BioRender account to export high-resolution images. Details on how to use and access the premium account are included as part of this email.

-Please include a full title page as part of your article (Word) file (containing title, authors, affiliations, corresponding author name and contact details, keywords, and running title).

EDITOR COMMENTS

Both reviewers deemed the study interesting and potentially impactful, but have several suggestions to strengthen the conclusions. These include:

- studying the impact of CaM mutations on KCNQ1 trafficking when KCNQ1 is expressed with KCNE1
- considering the effect of temperature
- attempting to estimate the contributions of other ion channels that are regulated by CaM

The authors are encouraged to address these important comments.

REFEREE COMMENTS

Referee #1:

This is a solid and comprehensive study that characterizes CaM mutant associated with LQTS, showing that the mutants do not bind properly to IKs channel and thereby impairing the calcium sensitivity and contributing to reduced repolarization reserve in patients.

My only concern is that all the voltage clamp studies are carried out at room temperature, and would be prudent to know if the same holds at 37°C degrees?

Since CaM mutants also affect other ionic currents (I_{Ca}, I_{Na}) that would also affect repolarization reserve, it would be appropriate to estimate quantitatively their contribution. There is no attempt to estimate how important IKs suppression is relative to other CaM effects in promoting LQTS. Perhaps this could be assessed with mathematical modeling. Comprehensive work.

Referee #2:

This is a very interesting paper with strong potential to provide important novel insight into the relationship between inherited mutations in calmodulin and generation of Long QT Syndrome arrhythmias caused by down regulation in an important potassium channel in heart, the IKs channel. There are functional studies in the figures that summarize this point, and the remainder of the study really focuses on the impact of key mutation on calmodulin structure. I have suggestions for additional experiments that I think will improve the conclusions drawn regarding the impact of calmodulin mutations on the IKs channel.

Shown in Figure 4 are clear experiments indicating that the CaM mutations studied do not affect KCNQ1 trafficking in a manner that would underlie the reduction in IKS channel illustrated in figures 2 and 3. This approach should be extended to carry out the same type of experiments studying the possible impact of these mutations on KCNQ1 trafficking when KCNQ1 is expressed with KCNE1 to form IKS channels. This would more directly address the important data shown in figures 2 and 3.

END OF COMMENTS

Confidential Review

08-May-2023

First of all, the authors would like to thank the reviewers for their work. It is appreciated that the reviewers acknowledged the amount of work undertaken to characterize the LQTS-associated CaM variants and that they find the data solid and interesting. This paper provides novel insights into the molecular mechanism of CaM-associated cardiac arrhythmias.

All comments are fair and insightful. All of the potential issues raised by the referees/editor have been addressed below.

EDITOR COMMENTS

Both reviewers deemed the study interesting and potentially impactful, but have several suggestions to strengthen the conclusions. These include:

- studying the impact of CaM mutations on KCNQ1 trafficking when KCNQ1 is expressed with KCNE1
 - considering the effect of temperature
 - attempting to estimate the contributions of other ion channels that are regulated by CaM
- The authors are encouraged to address these important comments.

REFEREE COMMENTS

Referee #1:

This is a solid and comprehensive study that characterizes CaM mutant associated with LQTS, showing that the mutants do not bind properly to IKs channel and thereby impairing the calcium sensitivity and contributing to reduced repolarization reserve in patients.

My only concern is that all the voltage clamp studies are carried out at room temperature, and would be prudent to know if the same holds at 37°C degrees?

Since CaM mutants also affect other ionic currents (I_{Ca} , I_{Na}) that would also affect repolarization reserve, it would be appropriate to estimate quantitatively their contribution. There is no attempt to estimate how important IKs suppression is relative to other CaM effects in promoting LQTS. Perhaps this could be assessed with mathematical modelling

Referee #2:

This is a very interesting paper with strong potential to provide important novel insight into the relationship between inherited mutations in calmodulin and generation of Long QT Syndrome arrhythmias caused by down regulation in an important potassium channel in heart, the IKs channel. There are functional studies in the figures that summarize this point, and the remainder of the study really focuses on the impact of key mutation on calmodulin structure. I have suggestions for additional experiments that I think will improve the conclusions drawn regarding the impact of calmodulin mutations on the IKs channel.

Shown in Figure 4 are clear experiments indicating that the CaM mutations studied do not affect KCNQ1 trafficking in a manner that would underlie the reduction in IKs channel illustrated in figures 2 and 3. This approach should be extended to carry out the same type of experiments studying the possible impact of these mutations on KCNQ1 trafficking when KCNQ1 is expressed with KCNE1 to form IKs channels. This would more directly address the important data shown in figures 2 and 3.

Response to the referees/editor

- *Kv7.1 surface density.* We have performed new experiments to determine the effect of CaM variants on the surface density of Kv7.1, when KCNE1 is present. The flow cytometry experiments did not show any significant effect of the LQTS-associated CaM variant on Kv7.1 surface density. Methods and Results have been updated accordingly.

Results

Trafficking of KCNQ1 channels to the plasma membrane is not altered by the LQTS CaM variants. [...] Based on this method developed by Colecraft's laboratory [1], we observed that KCNQ1 surface density (ECFP+Alexa Fluor 647 labelled) was $35.0 \pm 1.2\%$ (of total KCNQ1 channels, ECFP only, $n = 5$) and that LQTS-associated CaM variants did not significantly alter the % of channels at the plasma membrane. Kv7.1 surface density was $30.8 \pm 2.6\%$ for D95V ($n = 5$, $p = 0.3140$), $30.4 \pm 1.7\%$ ($n = 5$, $p = 0.2502$) for N97I and $35.4 \pm 1.9\%$ for D131H ($n = 5$, $p = 0.9976$) (Fig. 4).

Figure 4. Optical detection of cell surface density of Kv7.1 with flow cytometry. (a) Schematic showing Alexa Fluor 647 labelling of cell surface BBS-tagged ECFP- KCNQ1, KCNE1 and CaM-EGFP. HEK293T cells were co-transfected with ECFP-BBS-KCNQ1, KCNE1 and EGFP-labelled CaM at the C-terminus. A bungarotoxin binding-site (BBS) was introduced within the extracellular S1-S2 loop of KCNQ1, allowing for surface labelling using Alexa Fluor 647 conjugated to α -bungarotoxin (BTX₆₄₇). (b) Dot plots of surface-labelled Kv7.1 channels in HEK293T cells expressing ECFP-BBS-KCNQ1, KCNE1, CaM variants and incubated with α -bungarotoxin-Alexa Fluor 647. Plots show ECFP vs Alexa Fluor 647 fluorescence (arbitrary units). 10,000 cells were counted for each experiment. Dots represent a live, single cell as determined after SSC-A/FSC-A and FSC-H/FSC-A gating. Vertical and horizontal lines represent threshold values set based on isochronal and untransfected cells. Top left quadrant (blue) denotes ECFP-BBS-KCNQ1-expressing cells with little Alexa647 signal, indicating low channel surface density. Top right quadrant (purple) represents ECFP-BBS-KCNQ1-positive cells with robust channel trafficking to the

surface. Bottom quadrants indicate untransfected cells. (c) Analysis of flow cytometry data to determine the relative surface density for KCNQ1 in the presence of CaM variants (data has been filtered for EGFP-positive cells). Data were expressed as mean \pm s.e.m. Differences between groups were determined using a one-way ANOVA with Dunnett's multiple comparisons tests.

- *IKs measurement at 37deg.* While it would be interesting to determine the effect of temperature on IKs, this would be beyond the scope of this paper. For technical reasons, whole-cell patch clamp experiments on single isolated cells are usually conducted at RT, as evidenced by a large selection of papers investigating the effect of CaM variants on ion channel activity (in HEK293 cells, primary isolated cardiomyocytes, iPSCs) [2-8]. Therefore for direct comparison with the existing literature, we opted to perform our experiments at RT. This is the first study to define an effect of LQTS-associated CaM variants on IKs, and it will be important in future analysis to factor-in the effects of temperature.
- *Contribution of IKs using mathematical modelling.* Using a computer model of a ventricular action potential, we have estimated the effect of IKs alteration (using the data from this paper) on the action potential duration. We determine that the action potential was prolonged by 3.5-6.8%, which corresponds to a contribution of 9-13% to the LQTS phenotype. The modelling data has been included in this revised version of the manuscript. Methods, Results and Discussion have been updated accordingly.

Results

Predicted effect of IKs inhibition by CaM variants on action potential duration (APD).

First, based on our experimental data, we mathematically altered IKs to match the effect of the LQTS-associated CaM variants (Fig. 12a). Then, using the adapted version of the O'Hara – Rudy model, we simulated ventricular action potentials and measured the duration at 50% of the amplitude (APD50). APD50 increased from 216.8 ms (CaM-WT) to 224.7 ms (N97I), 226.7 ms (D131H) and 232.5 ms (D95V). This represents a relative increase of APD of 3.6% (N97I), 4.6% (D131H) and 7.2% (D95V), when compared to CaM-WT (Fig. 12b-d).

Discussion

[...] Using mathematical modelling, we predicted that IKs reduction for the LQTS CaM variants would cause an APD prolongation of 3.6% (N97I), 4.6% (D131H) and 7.2% (D95V). Considering that QTc interval is prolonged by 26% (N97I, from 440 ms to 555 ms) [9], 48% (D131H, from 440 ms to 651 ms) [10] and 57% (D95V, from 440 ms to 690 ms) [3], data suggest a substantial IKs contribution to the LQTS phenotype across all variants: 14% (N97I), 10% (D131H) and 13% (D95V).

References

1. Aromolaran, A.S., et al., *LQT1 mutations in KCNQ1 C-terminus assembly domain suppress IKs using different mechanisms*. Cardiovasc Res, 2014. **104**(3): p. 501-11.
2. Yin, G., et al., *Arrhythmogenic calmodulin mutations disrupt intracellular cardiomyocyte Ca²⁺ regulation by distinct mechanisms*. J Am Heart Assoc, 2014. **3**(3): p. e000996.
3. Pipilas, D.C., et al., *Novel calmodulin mutations associated with congenital long QT syndrome affect calcium current in human cardiomyocytes*. Heart Rhythm, 2016. **13**(10): p. 2012-9.
4. Boczek, N.J., et al., *Spectrum and Prevalence of CALM1-, CALM2-, and CALM3-Encoded Calmodulin Variants in Long QT Syndrome and Functional Characterization of a Novel Long QT Syndrome-Associated Calmodulin Missense Variant, E141G*. Circ Cardiovasc Genet, 2016. **9**(2): p. 136-146.
5. Gomez-Hurtado, N., et al., *Novel CPVT-Associated Calmodulin Mutation in CALM3 (CALM3-A103V) Activates Arrhythmogenic Ca Waves and Sparks*. Circ Arrhythm Electrophysiol, 2016. **9**(8).
6. Limpitikul, W.B., et al., *Calmodulin mutations associated with long QT syndrome prevent inactivation of cardiac L-type Ca(2+) currents and promote proarrhythmic behavior in ventricular myocytes*. J Mol Cell Cardiol, 2014. **74**: p. 115-24.
7. Limpitikul, W.B., et al., *A Precision Medicine Approach to the Rescue of Function on Malignant Calmodulinopathic Long-QT Syndrome*. Circ Res, 2017. **120**(1): p. 39-48.
8. Rocchetti, M., et al., *Elucidating arrhythmogenic mechanisms of long-QT syndrome CALM1-F142L mutation in patient-specific induced pluripotent stem cell-derived cardiomyocytes*. Cardiovasc Res, 2017. **113**(5): p. 531-541.
9. Crotti, L., et al., *Calmodulin mutations associated with recurrent cardiac arrest in infants*. Circulation, 2013. **127**(9): p. 1009-17.
10. Makita, N., et al., *Novel calmodulin mutations associated with congenital arrhythmia susceptibility*. Circ Cardiovasc Genet, 2014. **7**(4): p. 466-74.

Dear Dr Helassa,

Re: JP-RP-2023-284994R1 "Long QT syndrome-associated calmodulin variants disrupt the activity of the slowly activating delayed rectifier potassium channel (IKs)." by Liam F McCormick, Kirsty Wadmore, Amy Milburn, Nitika Gupta, Rachael Morris, Marie Held, Ohm Prakash, Joseph Carr, Richard Barrett-Jolley, Caroline Dart, and Nordine Helassa

We are pleased to tell you that your paper has been accepted for publication in The Journal of Physiology.

Authors should note that it is too late at this point to offer corrections prior to proofing. The accepted version will be published online, ahead of the copy edited and typeset version being made available. Major corrections at proof stage, such as changes to figures, will be referred to the Editors for approval before they can be incorporated. Only minor changes, such as to style and consistency, should be made at proof stage. Changes that need to be made after proof stage will usually require a formal correction notice.

Yours sincerely,

Natalia Trayanova
Senior Editor
The Journal of Physiology

P.S. - You can help your research get the attention it deserves! Check out Wiley's free Promotion Guide for best-practice recommendations for promoting your work at www.wileyauthors.com/eeo/guide. You can learn more about Wiley Editing Services which offers professional video, design, and writing services to create shareable video abstracts, infographics, conference posters, lay summaries, and research news stories for your research at www.wileyauthors.com/eeo/promotion.

IMPORTANT NOTICE ABOUT OPEN ACCESS: To assist authors whose funding agencies mandate public access to published research findings sooner than 12 months after publication, The Journal of Physiology allows authors to pay an Open Access (OA) fee to have their papers made freely available immediately on publication.

You can check if your funder or institution has a Wiley Open Access Account here: <https://authorservices.wiley.com/author-resources/Journal-Authors/licensing-and-open-access/open-access/author-compliance-tool.html>.

EDITOR COMMENTS

Reviewing Editor:

All comments have been addressed satisfactorily. Congratulations!

REFeree COMMENTS

Referee #1:

No further comments to add to my review of the previous version

Referee #2:

The authors have very nicely and completely addressed my previously-raised concerns and I have no additional and/or new concerns to address.

1st Confidential Review

15-Jun-2023